# Group Contrastive Learning for Weakly Paired Multimodal Data

## Abstract

We present GROOVE , a semi-supervised multi-modal representation learning approach for high-content perturbation data where samples across modalities are weakly paired through shared perturbation labels but lack direct correspondence. Our primary contribution is GroupCLIP, a novel group-level contrastive loss that bridges the gap between CLIP for paired cross-modal data and SupCon for uni-modal supervised contrastive learning, addressing a fundamental gap in contrastive learning for weakly-paired settings. We integrate GroupCLIP with an on-the-fly backtranslating autoencoder framework to encourage cross-modally entangled representations while maintaining group-level coherence within a shared latent space. Critically, we introduce a comprehensive combinatorial evaluation framework that systematically assesses representation learners across multiple optimal transport aligners, addressing key limitations in existing evaluation strategies. This framework includes novel simulations that systematically vary shared versus modality-specific perturbation effects enabling principled assessment of method robustness. Our combinatorial benchmarking reveals that there is not yet an aligner that uniformly dominates across settings or modality pairs. Across simulations and two real single-cell genetic perturbation datasets, GROOVE performs on par with or outperforms existing approaches for downstream cross-modal matching and imputation tasks. Our ablation studies demonstrate that GroupCLIP is the key component driving performance gains. These results highlight the importance of leveraging group-level constraints for effective multi-modal representation learning in scenarios where only weak pairing is available.

## 1 Introduction

Perturbation screens have gained major prominence in recent years for their ability to elucidate causal gene regulatory networks (Dixit et al., 2016), identify candidate therapeutic targets (Rood et al., 2024), and enable small molecule repurposing (Bhandari et al., 2022). Each given modality (e.g. RNA-Seq, ATAC-Seq, or high-content imaging) only observes a subset of the underlying biology of a system (Cui et al., 2025), therefore recent efforts have shifted toward multi-modal investigation of perturbation effects via paired profiling approaches (Frangieh et al., 2021; Martin-Rufino et al., 2025). While promising, this type of profiling remains feasible only for specific combinations of modalities, such as gene expression paired with chromatin accessibility (Martin-Rufino et al., 2025) or gene expression paired with surface protein measurements (Frangieh et al., 2021). Notably, it is not currently feasible to obtain both perturbed microscopy images from cell painting assays (Feldman et al., 2019) and perturbed gene expression profiles from the *same* individual cells, as both measurements are inherently destructive assays. In this setting, we do not have access to paired samples across modalities and can only broadly *group* cells by their perturbation (other experimental) *labels*.

Consequently, recent efforts in multi-modal perturbation screens have shifted toward developing computational approaches for post-hoc "pairing" (even though true pairs don't actually exist) of cells across modalities or cross-modal imputation (See Section 2). Both these objectives depend on learning a useful joint representation of the non-paired multi-modal data with group-level information only. Such a setting immediately rules out existing standard contrastive learning approaches for joint cross-modal inference (see Section 2). Cross-modal contrastive approaches like CLIP (Radford et al., 2021) need paired data while uni-modal label-based contrastive methods like SupCon (Khosla et al., 2020) are not natively compatible with multi-modal data. Moreover, existing multi-modal single-cell

deep learning approaches, such as uncoupled autoencoders (Samaran et al., 2024; Ashuach et al., 2023), rely on either strong human-defined priors to establish putative cell correspondences or access to paired data. No current framework can effectively learn from *native* weakly paired multi-modal data for a *well-mixed* multi-modal latent representation (See Section 2).

**Contributions.** In this work, we develop GROOVE (GROup cOntrastiVE learning for weakly paired multi-modal data) to address these challenges in weakly paired multi-modal data. Our approach makes three key contributions. First, we introduce a group-level semi-supervised contrastive loss, bridging the gap between CLIP (Radford et al., 2021) and SupCon (Khosla et al., 2020) for weakly paired multi-modal data. Second, GROOVE integrates this loss with *on-the-fly* backtranslating autoencoders adapted from neural machine translation (Artetxe et al., 2017), creating a unified architecture for learning from weakly paired single-cell data. Finally, we develop a comprehensive evaluation framework consisting of: (i) novel simulations that systematically vary the proportion of shared versus modality-specific information, and (ii) combinatorial benchmarking that pairs different representation learners with various alignment algorithms to assess both matching and cross-modal imputation performance.

**Notations.** In standard constructions of multi-modal learning, one works with a setting where all the data modalities are observed for all the samples such that $\mathcal{D} = \{(\boldsymbol{x}_i^{(1)}, \boldsymbol{x}_i^{(2)})\}_{i=1}^N$. However, the focus of this work is on settings where such paired data does not exist, i.e., we only have access to one data modality per sample. We instead have access to a common state, environment, intervention or *perturbation* label $t \in \mathcal{T} \subset \mathbb{Z}$ that is shared across modalities and samples. This additional information renders our data as *weakly paired* such that any data instance across the modalities without the same label $t$ are strictly unrelated. We can now re-formulate multi-modal learning in the *weakly paired* setting as having data $\mathcal{D}^{(m)}$ from two disjoint data modalities indexed by $m \in \mathbb{M} = \{1, 2\}$. Each modality-specific dataset is a collection of $N^{(m)}$ samples $\mathcal{D}^{(m)} = \{(\boldsymbol{x}_i^{(m)}, t_i)\}_{i=1}^{N^{(m)}}$, where each $\boldsymbol{x}_i^{(m)} \in \mathcal{X}^{(m)} \subseteq \mathbb{R}^{k^{(m)}}$ is the data instance for modality $m$ and its corresponding label $t_i$. Given this, our multimodal representation learning problem is to learning an embedding $\boldsymbol{z} \in \mathcal{Z} \subseteq \mathbb{R}^d$ for each sample in a *shared* low-dimensional representation space, such that $d \ll min(k^{(1)}, k^{(2)})$. And let $\mathcal{D}_z^{(m)} = \{(\boldsymbol{z}_i^{(m)}, t_i)\}_{i=1}^{N_m}$ represent the collection of latent representations and labels for modality $m$. We define $\bar{m}$ to denote the *other* modality.

## 2 BACKGROUND

In this section, we review related work from both the broader contrastive representation learning literature and the single-cell community, highlighting the gap our contributions aim to address. We also briefly review unsupervised neural translation which underpins our architecture.

**Contrastive representation learning.** Contrastive representation learning has emerged as a powerful general paradigm (Le-Khac et al., 2020), with theoretical underpinnings (Alshammari et al., 2025; Van Assel et al., 2025; HaoChen et al., 2021) and a wide range of practical instantiations. The foundational InfoNCE loss (Oord et al., 2018) maximizes mutual information between positives while minimizing it for negatives, and has since inspired numerous influential extensions (Chen et al., 2020; He et al., 2020; Grill et al., 2020). Despite their success, these methods suffer from a key limitation: they often misclassify samples from the same class as negatives. Supervised contrastive learning (Khosla et al., 2020) alleviates this by leveraging labels to form multiple positives per anchor, but remains uni-modal in design. Similarly, Yao et al. (2024) leverage perturbation labels to pair samples across batches, but operates at the group level as a uni-modal denoising approach. For multimodal learning, CLIP (Radford et al., 2021) and CMC (Tian et al., 2020) are the most influential approaches, aligning image–text pairs or extending contrastive objectives across multiple modalities. Other works, S-CLIP (Mo et al., 2023) and SemiCLIP (Gan et al., 2025), have explored supervised and semi-supervised extensions of CLIP that leverage *limited paired data* with additional label information. However, these methods still require some instance-level pairs between modalities and cannot operate in the purely weakly-paired regime where only group labels connect modalities. Their reliance on strictly paired data remains a critical bottleneck. Thus, a key gap in the literature is the absence of a supervised extension of CLIP that can exploit weak pairing (Figure 1a).

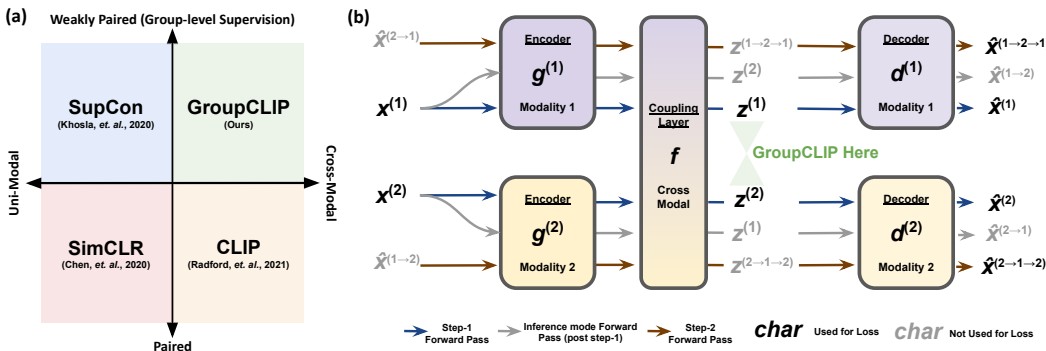

Figure 1: **(a)** GroupCLIP in the context for broader contrastive learning can be viewed as the multi-modal generalization of SupCon. **(b)** GROOVE Architecture and training step illustration. Each iteration consists of two steps: (1) optimize reconstruction and GroupCLIP losses, then (2) generate cross-modal pseudo-samples in inference mode and optimize the backtranslation loss.

**Weakly paired learning for multimodal single-cell data.**    Single-cell data is inherently unpaired because measurements are destructive, preventing multiple modalities from being captured from the same cell. This limitation has motivated the development of computational methods for unpaired multimodal integration (Yang et al., 2021; Samaran et al., 2024) and weakly paired learning (Xi et al., 2024; Ryu et al., 2025), particularly in settings involving perturbations, which are the focus of our work. A particularly important downstream application of these methods is cross-modal imputation, where one modality is predicted from observations of another. Next, we briefly describe the most relevant works tackling this problem. First, Xi et al. (2024) leverage perturbation labels via propensity score matching. Their method trains independent classifiers to predict perturbations in each modality and then uses the resulting logits to define a common support for alignment, following the classical balancing scores of Rubin (1974). Although conceptually appealing, this approach has notable limitations: it assumes that all perturbation-induced variation is perfectly shared across modalities, ignoring modality-specific effects; and the learned latent representations (classifier logits) capture only perturbation-predictive information, discarding potentially valuable *intra*-perturbation variation that may be critical for downstream tasks. Second, Ryu et al. (2025) address the alignment stage of the pipeline by introducing a label-constrained variant of the Gromov–Wasserstein optimal transport (GW-OT) problem (Mémoli, 2011). Unlike approaches that rely on direct sample-wise correspondences, GW-OT aligns representations by minimizing structural discrepancies between the metric spaces induced by the two modalities (Sebbouh et al., 2024; Van Assel et al., 2024). This formulation allows alignment across modality-specific latent spaces of different dimensions, each obtained via PCA. The method's effectiveness, however, is constrained by (i) the quality and robustness of the latent representations, (ii) the cubic computational complexity of GW-OT (Peyré et al., 2016), and (iii) its non-convexity, which makes it susceptible to local minima (Vayer, 2020). Finally, Samaran et al. (2024) propose the only method that integrates alignment and imputation within a unified framework. Yet, their approach requires a predefined set of aligned features shared across modalities—a strong assumption that is rarely satisfied in practice—and it does not leverage perturbation labels that could substantially enhance alignment.

**Multi-lingual Neural Translation.**    Neural machine translation has demonstrated that high-quality translation systems can be trained using only unpaired monolingual data through unsupervised approaches that employ shared encoder architectures, dual training objectives combining reconstruction and back-translation losses, and cross-lingual initialization strategies (Sennrich et al., 2015; Artetxe et al., 2017; Lample et al., 2018; Conneau et al., 2018). Central to these methods is an *on-the-fly* backtranslation strategy designed to encourage entangled, cross-modally[1] informative latent representations. During training, a sample from one modality is encoded to the shared latent space, then this representation is used to generate a corresponding translation in the *other* modality. Parameters are updated via a two step optimization procedure based on a *reconstruction* (Equation 1) and *backtranslation* (Equation 2) loss. The reconstruction ($\hat{x}_i^{(m)}$) loss computes the error in encoding and decoding $x_i^{(m)}$ from the same modality and the backtranslation loss computes the error when

---

[1] We use modal and lingual interchangeably here.

encoding $\boldsymbol{x}_i^{(m)}$ and decoding to the other modality ($\hat{\boldsymbol{x}}_i^{(m \to \bar{m})}$), followed by encoding and decoding $\hat{\boldsymbol{x}}_i^{(m \to \bar{m})}$ back to it's original modality. This backtranslation strategy encourages the latents to be well-mixed across modalities. And as the model improves higher-quality pseudo-pairs are generated, creating a positive feedback loop that enhances both cross-modal alignment and within-modality representations. See Appendix A.1 for extended details on this procedure and the architecture. To the best of our knowledge, iterative back-translation from multilingual neural machine translation has not yet been applied to weakly paired single-cell multi-modal data.

$$\mathcal{L}_{\text{reconstruction}} = \frac{1}{|\mathbb{M}|} \sum_{m \in \mathbb{M}} \frac{1}{|\mathcal{D}^{(m)}|} \sum_{\boldsymbol{x}_i^{(m)} \in \mathcal{D}^{(m)}} \left\| \hat{\boldsymbol{x}}^{(m)} - \boldsymbol{x}_i^{(m)} \right\|_2^2 \tag{1}$$

$$\mathcal{L}_{\text{backtranslation}} = \frac{1}{|\mathbb{M}|} \sum_{m \in \mathbb{M}} \frac{1}{|\mathcal{D}^{(m)}|} \sum_{\boldsymbol{x}_i^{(m)} \in \mathcal{D}^{(m)}} \left\| \hat{\boldsymbol{x}}^{(m \to \bar{m} \to m)} - \boldsymbol{x}_i^{(m)} \right\|_2^2 \tag{2}$$

## 3 PROPOSED METHOD

We instantiate our base architecture using uncoupled autoencoders, which are standard in the single-cell literature (Samaran et al., 2024; Lopez et al., 2018), with an added shared linear (coupling) projection layer across modalities. We train using the two-step optimization procedure from unsupervised machine translation (Appendix A for more details). This base model, however, does not leverage the additional supervisory signal present in the sample associated (perturbation) labels $\mathcal{T}$. We can make use of this additional information to increase the level of supervision of our *on-the-fly* autoencoder from unsupervised to semi-supervised. The label information allows us to update our latent representation with the following desiderata: (1) sample across modalities with the same label should be pushed close together and (2) any samples without the same label should be as far away as possible (repulsed). Such a formulation invokes contrastive learning as a natural solution. This motivates us to develop a novel contrastive loss for weakly paired multi-modal data. Our approach leverages the weak pairing structure by treating samples from different modalities that share the same label as positive pairs. For a latent representation $\boldsymbol{z}_i^{(m)}$ from modality $m$, we define its attractors as all latent representations from the other modality $\bar{m}$ that share the same label.

Formally, we define $\mathcal{P}_i^{(m)} = \{ \boldsymbol{z}_j^{(\bar{m})} \in \mathcal{D}_z^{(\bar{m})} : t_j = t_i \}$ as the collection of *positive* samples for anchor $\boldsymbol{z}_i^{(m)}$ (same label, opposite modality) and $\mathcal{A}_i^{(m)} = \mathcal{D}_z^{(\bar{m})}$ as the collection of all *candidates* from the other modality (includes both positives and negatives). Following CLIP (Radford et al., 2021), we normalize over *all* candidates from the other modality (including positives). Given the anchor $\boldsymbol{z}_i^{(m)}$, the loss is:

$$\ell_i^{(m)} = -\log \frac{\sum\limits_{\boldsymbol{z}_p \in \mathcal{P}_i^{(m)}} \exp\big(\text{sim}(\boldsymbol{z}_i^{(m)}, \boldsymbol{z}_p)/\tau\big)}{\sum\limits_{\boldsymbol{z}_a \in \mathcal{A}_i^{(m)}} \exp\big(\text{sim}(\boldsymbol{z}_i^{(m)}, \boldsymbol{z}_a)/\tau\big)} \tag{3}$$

where $\text{sim}(\boldsymbol{u}, \boldsymbol{v})$ is a similarity function between two embeddings (such as cosine similarity) and $\tau > 0$ is the temperature parameter used to scale similarities. Note that in addition to the standard cosine similarity, we also experimented with *t*-distribution based similarity kernels, see Appendix A.4. Averaging over all anchors and modalities gives:

$$\mathcal{L}_{\text{GroupCLIP}} = \frac{1}{|\mathbb{M}|} \sum_{m \in \mathbb{M}} \frac{1}{|\mathcal{D}_z^{(m)}|} \sum_{\boldsymbol{z}_i^{(m)} \in \mathcal{D}_z^{(m)}} \ell_i^{(m)} \tag{4}$$

The resulting GroupCLIP loss encourages latent representations of the same label to cluster together across modalities while pushing apart representations of all other labels. Because contrastive learning is sensitive to batch composition, we employ a balanced under-sampling strategy that maintains equal sample counts per label in each mini-batch, preventing class imbalance without oversampling rare labels, see Appendix A.5.

Our approach fills a crucial gap in the contrastive learning landscape by extending supervised contrastive learning to the cross-modal setting (Figure 1a). Just as CLIP (Radford et al., 2021) represents the canonical cross-modal extension of SimCLR's (Chen et al., 2020) unsupervised contrastive framework, GroupCLIP serves as the natural cross-modal extension of SupCon's (Khosla et al., 2020) supervised contrastive approach. While CLIP leverages naturally paired data without explicit label supervision, GroupCLIP leverages label supervision without requiring natural pairings between modalities. This distinction is critical for biological applications where true cross-modal pairs are often unavailable, but perturbation labels provide rich supervisory signal. By bringing supervised contrastive learning to the multi-modal domain, GroupCLIP bridges the gap between uni-modal supervised methods and cross-modal unsupervised approaches, offering a principled framework for scenarios with weak pairing but strong label information.

Finally, by integrating our novel GroupCLIP loss (Equation 4) with the *on-the-fly* backtranslating autoencoder framework (Equation 1,2), we develop the GROOVE architecture – a novel method that learns a unified representations from weakly paired multi-modal data with the following losses:

$$\mathcal{L}_{\text{step-1}} = \alpha \cdot \mathcal{L}_{\text{GroupCLIP}} + \beta \cdot \mathcal{L}_{\text{reconstruction}} \quad (5) \qquad \mathcal{L}_{\text{step-2}} = \beta \cdot \mathcal{L}_{\text{backtranslation}} \quad (6)$$

$\alpha, \beta$ are hyperparameter that balance the loss components. Algorithm 1 in Appendix A.6 sketches the overall training loop of GROOVE and Figure 1b is a visual illustration of this architecture.

## 4 EVALUATION FRAMEWORK

We evaluate the quality of our learned latent representations through two approaches. First, we assess their utility for OT-based cross-modal sample matching, which is a standard evaluation (Ryu et al., 2025; Xi et al., 2024) since high-quality latent representations should enable the recovery of meaningful transport plans that correctly identify and match similar samples across modalities. We then evaluate performance on a key downstream task leveraging the transport plan: uni-directional imputation, where we predict one modality given samples from another.

### 4.1 COMBINATORIAL EVALUATION

We identify a key limitation in previous evaluations: they either hold the representation learner fixed while comparing various OT methods (Ryu et al., 2025), or fix the OT approach while comparing different representation learners (Xi et al., 2024). This evaluation paradigm can produce systematically biased results because the performance of representation learning methods is inherently coupled with the choice of downstream alignment algorithm. The coupling arises from fundamental differences in the geometric assumptions underlying various OT formulations.

For instance, Entropic Optimal Transport (EOT) implicitly assumes that both modalities can be embedded into a shared feature space, such that cross-modal correspondences are meaningful when representations are compared using the same metric. In other words, the learned embeddings are expected to preserve distances consistently across modalities in a common coordinate system. By contrast, Entropic Gromov–Wasserstein Optimal Transport (EGWOT) does not assume a shared space; instead, it treats the modalities as potentially distinct metric spaces and aligns them by comparing their internal distance structures rather than absolute coordinates. These differing assumptions suggest that representation learners optimized for one geometric framework may not perform optimally under alternative OT formulations. To address this limitation, we propose a combinatorial evaluation framework that systematically tests each representation learner against all available OT aligners. This approach allows us to assess how representation learners and alignment algorithms interact, yielding more robust performance evaluations that account for the sensitivity of downstream choices.

### 4.2 BASELINES

We focus our evaluation on approaches that can natively operate on weakly-paired data without requiring pre-specified feature correspondences or paired samples. This includes the following representation learning baselines: Propensity Score (PS) matching (Xi et al., 2024) and domain-adversarial variational autoencoder with label adaptation (DAVAE), a custom modification we made of Ashuach et al. (2023) as described in Ryu et al. (2025). PS represents the most recent approach

for weakly paired cross-modal matching and is the only existing method that directly addresses this problem in its native form. DAVAE is conceptually closest to our proposed approach but differs in employing modality identification adversarial loss and using linear probe supervision for latent space regularization. All methods use identical architectures with consistent latent embedding dimensions, and we evaluate each representation learner in conjunction with both standard (EOT, EGWOT) and label-constrained optimal transport approaches (LabeledEOT, LabeledEGWOT, labeledCOOT). See Appendix B for details.

For unidirectional imputation, we trained a 2-layer MLP interleaved with ReLU activations and a final linear projection layer. This model was trained using the transport plan returned by the OT aligner which defines the training sampling strategy. Specifically, when training to predict modality 1 from modality 2 and given a transport plan $T \in [0, 1]^{N^{(1)} \times N^{(2)}}$, we sample index-$j$ from the modality 1 for each sample index-$i$ from modality 2 as function of $j \sim \text{Multinomial}(\frac{T_{:,i}}{\sum T_{:,i}})$, for each mini-batch.

## 4.3 DATASETS

**Simulations.** A key limitation in previous evaluations is the assumption that variation from perturbations are fully shared between modalities, making modality-specific variation completely independent of labels, an unrealistic assumption for biological data (Argelaguet et al., 2020; Lin & Zhang, 2023). To address this, we developed a probabilistic simulation framework that captures both shared and modality-specific latent variation affected by perturbations. Our framework models each cell through shared latent factors (affecting both modalities identically) and modality-specific factors (unique to each molecular layer), with two perturbation types: shared perturbations affecting both modalities jointly, and modality-specific perturbations targeting individual modalities independently. We systematically evaluate method performance across different coupling levels by varying the proportion of shared versus modality-specific dimensions, testing scenarios with 100%, 80%, and 50% shared variation. Full simulation details are provided in Appendix C. For each scenario, we simulate 10 replicates with perturbation-balanced 80-20 train-test splits for evaluation.

**Perturb-Multiome** We use the paired gene expression and chromatin accessibly (multiome) dataset from Martin-Rufino et al. (2025), consisting of transcription factor based perturbations for a more realistic evaluation. The original data was pre-processed and subset to result in 2560 cells over 20 perturbation with 128 cells per perturbation with 512 genes and 256 gene accessibility scores. See details in Appendix D. This dataset was analyzed under two frameworks: (1) perturbation balanced 5-fold splits, and (2) leave-one-perturbation-out (LOPO). All imputation results for this dataset focus on predicting gene expression from accessibility scores.

**Perturb-CITE-seq.** We also use the Frangieh et al. (2021) Pertub-CITE-seq data, consisting of genetic perturbation with paired RNA-seq and surface protein measurements, for a more realistic evaluation. The original data was pre-processed and subset to result in 3689 cells over 19 genetic perturbation (cells per condition: median [min, max] = 201 [72, 270]) with 20 proteins and 500 genes, which were directly used as input. See details in Appendix E. This dataset was also analyzed under balanced 5-fold splits and LOPO cross-validation. All imputation results for this dataset focus on predicting gene expression from protein level measurements.

## 4.4 METRICS

Since our datasets contain true cell pairings, we assess OT matching accuracy using two ground-truth-based metrics. The *trace* of the normalized transport plan measures the proportion of true pairs being perfectly matched (Xi et al., 2024), e.g., it is 1.0 when all weight is assigned to true pairs and $1/N$ for uninformative uniform assignments. The symmetric Barycentric Fraction of Cells Closer Than True Match (*Bary. FOSCTTM*) measures how much weight the transport plan incorrectly assigns to false pairs relative to true pairs (Demetci et al., 2022; Liu et al., 2019), where 0.0 indicates perfect matching and 0.5 represents random uniform assignment. We report the symmetric Bary. FOSCTTM by averaging performance across both modalities. Next, we quantify imputation accuracy using 6 metrics: *MSE*, 1-Wasserstein distance (*WD*), Cosine Similarity (*Cos-sim*), *KNN Recall*, KNN average precession-recall (*KNN PR*) and KNN area under the receiver-operating characteristic curve (*KNN ROC*). MSE and WD are standard metrics in the imputation literature (Gorla et al., 2025), where

Table 1: Matching performance metrics for top 5 method combinations in each shared proportion settings in simulations. SEs follow $\pm$; best in bold, second-best underlined.

| Shared Prop. | Method | Mean Rank | Trace | Bary. FOSCTTM |
|---|---|---|---|---|
| 100 | GROOVE (cosine)+ LabeledCOOT | 1.5 | **0.856±0.027** | 0.027±0.006 |
| | GROOVE (cosine)+ LabeledEOT | 3.0 | 0.466±0.020 | **0.026±0.004** |
| | DAVAE+ LabeledCOOT | 3.5 | 0.669±0.034 | 0.066±0.009 |
| | DAVAE+ LabeledEOT | 4.5 | 0.453±0.023 | 0.042±0.006 |
| | PS+LabeledEOT | 5.5 | 0.366±0.006 | 0.054±0.006 |
| 80 | DAVAE+ LabeledEOT | 2.5 | 0.165±0.008 | **0.146±0.014** |
| | GROOVE (cosine)+ LabeledCOOT | 3.5 | **0.237±0.015** | 0.180±0.012 |
| | GROOVE (cosine)+ LabeledEOT | 4.5 | 0.148±0.008 | 0.150±0.013 |
| | PS+LabeledEOT | 4.5 | 0.156±0.005 | 0.161±0.011 |
| | DAVAE+ LabeledCOOT | 6.0 | 0.202±0.014 | 0.196±0.011 |
| 50 | DAVAE+ LabeledEOT | 2.5 | 0.125±0.007 | **0.162±0.006** |
| | GROOVE (cosine)+ LabeledEOT | 4.0 | 0.118±0.007 | **0.162±0.009** |
| | GROOVE (cosine)+ LabeledCOOT | 5.0 | **0.184±0.015** | 0.206±0.008 |
| | GROOVE (tdist)+ LabeledEOT | 5.5 | 0.105±0.006 | 0.168±0.009 |
| | DAVAE+ LabeledCOOT | 6.0 | 0.164±0.011 | 0.214±0.007 |

the former assess global error and the latter quantifies the distributional alignment. Cos-sim is a common metric in perturbation prediction tasks (Littman et al., 2025; Adduri et al., 2025). The KNN-based metrics are motivated by recent work from (Littman et al., 2025), which demonstrated that KNN metrics provide robust assessment of local similarity preservation in perturbation effect prediction tasks. KNN metrics avoid bias from single gene failures by evaluating neighborhood preservation rather than global prediction errors. All metrics are reported on held-out data (either test sets or cross-validation folds) with standard errors (SE; across folds or replicates). Standard errors are rounded to three decimal places; values reported as 0.000 indicate SE< 0.0005. For further details on metrics, see Appendix F.

## 5 RESULTS

### 5.1 OVERALL SIMULATED DATA PERFORMANCE

Table 1 presents the matching performance of the top-5 ranked methods across different shared proportion settings in simulated data. Under the standard 100% shared perturbation-relevant variation setting, GROOVE (combined with LabeledCOOT) achieves superior performance compared to all other method combinations. At 80% and 50% shared proportions, GROOVE ranks second on average; however, it maintains the highest trace-based matching performance across all shared proportion conditions. Since matching performance serves as an intermediate step toward downstream objectives, Table 2 evaluates downstream imputation performance in the same simulated data. GROOVE -based approaches again achieve the highest rankings under both 100% and 80% shared conditions, while a PS-based combination is ranked marginally higher under 50% shared conditions. Statistical tests (Appendix G) confirm GROOVE achieves significant performance gains (p<0.05) under 100% shared conditions across both matching and imputation tasks, with cross metric average win rates competitive with the next-highest ranking methods in other settings. A closer inspection of the metrics reveals that GROOVE -based combinations consistently achieves top performance on nearly every individual metric except KNN PR, where they trail PS+LabeledEOT by 0.001 (much less than SE). Comparing the rankings between Tables 1 and 2, we notice that better matching performance does not guarantee optimal downstream task performance. For instance, DAVAE demonstrates better average matching performance for shared proportion $< 100\%$, but PS-based combinations yield better imputation performance under the same conditions. Despite these variations, GROOVE -based combination deliver top performance under 100% shared variation and demonstrates robust, consistent performance, securing *atleast* top-two ranks across all evaluated scenarios.

### 5.2 MULTIMODAL SINGLE-CELL PERFORMANCE

We now evaluate performance on the more realistic perturb-CITE-seq dataset. Table 3 presents imputation performance for the top-10 method combinations under 5-fold cross-validation. In contrast to the simulated data results, GROOVE -based approaches occupy the top-5 ranks while

Table 2: Imputation performance metrics for top 5 method combinations in each shared proportion settings in simulations. SEs follow ±; best in bold, second-best underlined.

| Shared Prop. | Method | Mean Rank | MSE | Cos-sim | KNN Recall | KNN PR | KNN ROC | WD |
|---|---|---|---|---|---|---|---|---|
| 100 | GROOVE (cosine)+LabeledCOOT | 1.00 | **0.099±0.012** | **0.949±0.005** | **0.444±0.029** | **0.272±0.024** | **0.706±0.015** | **0.065±0.003** |
| | DAVAE+LabeledCOOT | 2.00 | 0.121±0.013 | 0.931±0.006 | 0.412±0.027 | 0.244±0.021 | 0.689±0.014 | 0.092±0.005 |
| | PS+LabeledCOOT | 3.50 | 0.148±0.012 | 0.920±0.006 | 0.382±0.023 | 0.221±0.017 | 0.673±0.012 | 0.107±0.004 |
| | GROOVE (tdist)+LabeledCOOT | 4.00 | 0.146±0.014 | 0.931±0.004 | 0.372±0.020 | 0.216±0.014 | 0.668±0.011 | 0.097±0.005 |
| | DAVAE+LabeledEOT | 4.67 | 0.156±0.010 | 0.916±0.005 | 0.379±0.021 | 0.220±0.015 | 0.672±0.011 | 0.142±0.004 |
| 80 | GROOVE (cosine)+LabeledEOT | 2.83 | 0.623±0.119 | **0.837±0.011** | **0.241±0.014** | **0.137±0.008** | 0.598±0.008 | 0.199±0.027 |
| | PS+LabeledEOT | 3.00 | 0.593±0.115 | 0.836±0.010 | 0.239±0.013 | 0.135±0.007 | 0.597±0.007 | **0.198±0.027** |
| | GROOVE (cosine)+LabeledCOOT | 5.00 | 0.740±0.197 | 0.832±0.016 | 0.240±0.017 | 0.134±0.008 | **0.598±0.009** | 0.126±0.010 |
| | PS+LabeledEGWOT | 5.00 | 0.614±0.113 | 0.834±0.011 | 0.238±0.014 | 0.135±0.007 | 0.597±0.007 | 0.256±0.028 |
| | PS+EOT | 5.33 | **0.601±0.111** | 0.832±0.010 | 0.236±0.014 | 0.134±0.007 | 0.596±0.007 | 0.187±0.022 |
| 50 | PS+LabeledEOT | 3.83 | 0.516±0.040 | 0.830±0.011 | **0.215±0.011** | **0.123±0.006** | 0.585±0.006 | 0.183±0.011 |
| | GROOVE (cosine)+EOT | 4.00 | 0.519±0.039 | **0.838±0.009** | **0.215±0.009** | 0.122±0.006 | **0.585±0.005** | 0.208±0.012 |
| | GROOVE (cosine)+LabeledEOT | 4.00 | **0.513±0.040** | 0.835±0.011 | 0.212±0.010 | 0.121±0.006 | 0.583±0.005 | 0.194±0.012 |
| | DAVAE+LabeledEOT | 5.33 | 0.517±0.041 | 0.833±0.009 | 0.211±0.012 | 0.121±0.007 | 0.582±0.006 | 0.189±0.009 |
| | GROOVE (cosine)+LabeledCOOT | 6.00 | 0.559±0.049 | 0.832±0.012 | 0.213±0.010 | 0.121±0.006 | 0.584±0.005 | **0.134±0.009** |

PS-based approaches rank in the bottom two positions. This pattern supports the robustness of GROOVE -based combinations across diverse experimental conditions. Next, Table 8 (Appendix H) further shows that GROOVE -based approaches achieve the highest matching performance rankings. Notably, a PS-based approach ties with GROOVE -based methods for top matching performance; however, consistent with our simulation findings, superior matching performance does not necessarily translate to improved downstream imputation results. We further report performance of the top-10 methods under LOPO cross-validation, with matching and imputation performance results presented in Tables 9 and 10 (Appendix H), respectively. Even under this setting, GROOVE -based methods take the top-2 ranks in imputation. We also attain the single highest Trace and Bary. FOSCTTM metrics.

We next evaluate performance on the perturb-Multiome dataset. Tables 14 and 12 (Appendix I) summarize matching and imputation results, respectively, for the top 10 methods under five-fold cross-validation. Consistent with previous datasets, GROOVE attains the highest overall rank in both tasks. These sets of results in conjunction with the simulations indicates that cosine similarity kernel is often a good default but there are situations where the $t$-distribution kernel is helpful. We also again observe that high matching performance does not necessitate good imputation performance; DAVAE's better matching performance compared to PS-based approaches, yet DAVAE fails to rank within the top-10 for imputation performance. Similarly, GROOVE -based methods occupy the the top-3 ranks of matching (Appendix I, Table. 13) and imputation performance under LOPO evaluation. Interestingly, EOT variants enable superior imputation performance in the perturb-Multiome dataset when holding the underlying representation learner fixed, while EGWOT variants (of which COOT is a member) are preferred for imputation tasks in the perturb-CITE-seq dataset. This pattern suggests that optimal OT aligner selection depends on dataset characteristics, including the specific modality pairs, independent of the chosen representation learner.

Table 3: Imputation performance metrics for top 10 method combinations in Perturb-CITE-seq dataset with 5-fold evaluation. SEs follow ±; best in bold, second-best underlined, homogeneous metrics unannotated.

| Method | Mean Rank | MSE | Cos-sim | KNN Recall | KNN PR | KNN ROC | WD |
|---|---|---|---|---|---|---|---|
| GROOVE (cosine)+ LabeledEGWOT | 6.83 | **0.261±0.001** | 0.049±0.002 | 0.020±0.001 | 0.017±0.000 | 0.503±0.000 | 0.353±0.001 |
| GROOVE (cosine)+ LabeledCOOT | 7.33 | 0.282±0.000 | 0.016±0.002 | **0.021±0.001** | 0.017±0.000 | 0.503±0.000 | 0.297±0.000 |
| GROOVE (tdist)+ LabeledCOOT | 7.50 | 0.282±0.001 | 0.018±0.001 | 0.020±0.001 | 0.017±0.000 | 0.503±0.000 | 0.297±0.000 |
| GROOVE (tdist)+ LabeledEGWOT | 7.67 | **0.261±0.000** | 0.046±0.002 | 0.019±0.001 | 0.017±0.000 | 0.502±0.000 | 0.353±0.001 |
| GROOVE (tdist)+ LabeledEOT | 7.67 | 0.262±0.000 | 0.042±0.002 | 0.020±0.001 | 0.017±0.000 | 0.503±0.000 | 0.347±0.002 |
| DAVAE+ LabeledEGWOT | 8.00 | **0.261±0.000** | **0.059±0.002** | 0.018±0.001 | 0.017±0.000 | 0.502±0.000 | 0.352±0.000 |
| DAVAE+ LabeledCOOT | 8.33 | 0.295±0.000 | 0.019±0.001 | 0.020±0.001 | 0.017±0.000 | 0.503±0.001 | **0.281±0.000** |
| GROOVE (cosine)+ LabeledEOT | 9.00 | 0.262±0.001 | 0.044±0.002 | 0.019±0.001 | 0.017±0.000 | 0.502±0.000 | 0.350±0.001 |
| PS+ LabeledCOOT | 9.67 | 0.295±0.001 | 0.018±0.002 | 0.019±0.001 | 0.017±0.000 | 0.502±0.000 | **0.281±0.001** |
| PS+ LabeledEOT | 10.00 | 0.262±0.001 | 0.040±0.001 | 0.019±0.001 | 0.017±0.000 | 0.502±0.000 | 0.344±0.001 |

Table 4: Imputation performance metrics for top 10 method combinations in Perturb-Multiome dataset with leave one perturbation out evaluation. SEs follow $\pm$; best in bold, second-best underlined.

| Method | Mean Rank | MSE | Cos-sim | KNN Recall | KNN PR | KNN ROC | WD |
|---|---|---|---|---|---|---|---|
| GROOVE (tdist)+ LabeledEOT | 2.17 | **0.306±0.003** | **0.140±0.015** | 0.192±0.007 | 0.132±0.004 | 0.558±0.004 | 0.425±0.003 |
| GROOVE (cosine)+ LabeledEOT | 4.83 | 0.310±0.002 | 0.078±0.008 | **0.198±0.008** | **0.133±0.004** | **0.561±0.004** | 0.432±0.001 |
| GROOVE (tdist)+ EOT | 5.00 | 0.309±0.002 | 0.100±0.012 | 0.180±0.007 | 0.126±0.003 | 0.551±0.005 | 0.431±0.002 |
| PS+ LabeledEOT | 6.67 | 0.310±0.002 | 0.074±0.008 | 0.177±0.007 | 0.122±0.003 | 0.550±0.004 | 0.431±0.002 |
| GROOVE (cosine)+ EOT | 7.17 | 0.311±0.002 | 0.073±0.006 | 0.183±0.006 | 0.127±0.003 | 0.553±0.004 | 0.433±0.001 |
| GROOVE (cosine)+ LabeledCOOT | 7.67 | 0.357±0.014 | 0.072±0.030 | 0.170±0.008 | 0.122±0.003 | 0.546±0.004 | 0.326±0.004 |
| DAVAE+ LabeledEOT | 8.33 | 0.310±0.003 | 0.080±0.004 | 0.117±0.006 | 0.101±0.002 | 0.517±0.003 | 0.428±0.002 |
| PS+ EOT | 9.50 | 0.310±0.002 | 0.077±0.008 | 0.129±0.006 | 0.104±0.002 | 0.524±0.003 | 0.432±0.002 |
| GROOVE (tdist)+ LabeledCOOT | 10.33 | 0.363±0.013 | 0.050±0.030 | 0.161±0.008 | 0.119±0.004 | 0.541±0.004 | **0.324±0.005** |
| DAVAE+ LabeledCOOT | 10.50 | 0.344±0.010 | 0.062±0.023 | 0.148±0.007 | 0.113±0.003 | 0.534±0.004 | 0.348±0.004 |

## 5.3 ABLATION ANALYSIS

We conduct ablation analyses to assess the relative importance of key components within the GROOVE approach. These analyses fix the similarity kernel to cosine similarity and the aligner. We examine two ablation configurations: (1) 'No GroupCLIP': uses *on-the-fly* backtranslation with reconstruction and backtranslation losses but removes the GroupCLIP loss, and (2) 'Autoencoder only': uses standard reconstruction loss without GroupCLIP or backtranslation. We evaluate both configurations under the 80% shared proportion simulation setting (with LabeledEOT) and on the real Perturb-Multiome dataset (with EOT). Table H presents the ablation results. Across both simulated and real data, removing the GroupCLIP loss results in the largest performance decrease across all metrics. While both ablations show substantial degradation relative to full GROOVE , the difference between standard autoencoder and backtranslation (without GroupCLIP) is smaller by comparison, underscoring that GroupCLIPis the primary driver of performance gains. The additional supervision provided by GroupCLIP adds meaningful soft constraints to the latent representation. This supports our hypothesis that backtranslation alone is insufficient for encouraging group-level coherence and cross-group discrimination.

In the Pertub-Multiome dataset (under five-fold cross-validation), we see a meaningful increase accross most metrics by adding *on-the-fly* backtranslation to the standard autoencoder framework. However, in simulations, there difference between these two approaches is minimal. We see two possible, but not mutually exclusive, explanations for this dependency. First, our simulations still do not capture the full complexity of real multi-modal single-cell data that backtranslation can leverage. Second, the original *on-the-fly* backtranslation framework (Artetxe et al., 2017) utilized a shared, pre-trained multi-modal encoder, which was not available within the scope of this work. These factors may jointly or independently lead to underestimating both the utility of backtranslation and GROOVE overall potential in simulations. Note that we try to performe ablations in the Pertub-CITE-seq dataset (Appendix H), but the metrics showed insufficient dynamic range to support meaningful conclusions.

We have additionally conducted a hyperparameter sweep to quantitatively assess sensitivity to key parameters. Figure 2 presents contour plots of average matching performance (Trace and Bary. FOSCTTM) across the 100%, 80%, and 50% shared proportion settings as functions of temperature ($\tau$) and and reconstruction/backtranslation weight ($\beta$). The results reveal that GROOVE is relatively robust to $\tau$ but more sensitive to $\beta$. As with all deep learning approaches, we expect the hyperparameter sensitivity landscape to vary substantially across datasets and objectives. We recommend users perform dataset-specific hyperparameter optimization for best results.

Table 5: GROOVE ablation analysis performance metrics under 80% shared variation simulations and Perturb-Multiome dataset. SEs follow $\pm$; best in bold, second-best underlined.

| Dataset | Abation Type | Bary. FOSCTTM | MSE | Cos-sim | KNN Recall | KNN PR | KNN ROC | WD |
|---|---|---|---|---|---|---|---|---|
| Perturb-Multiome | GROOVE (cosine) | **0.489±0.007** | **0.308±0.003** | **0.102±0.028** | **0.044±0.007** | **0.028±0.002** | **0.512±0.004** | **0.428±0.005** |
| | No GroupCLIP | 0.497±0.001 | 0.310±0.001 | 0.076±0.007 | 0.036±0.002 | 0.025±0.000 | 0.507±0.001 | 0.432±0.002 |
| | Autoencoder only | 0.500±0.000 | 0.311±0.001 | 0.069±0.002 | 0.029±0.002 | 0.025±0.000 | 0.504±0.001 | 0.433±0.001 |
| Simulations | GROOVE (cosine) | **0.143±0.012** | **0.622±0.122** | **0.836±0.011** | **0.239±0.015** | **0.137±0.008** | **0.597±0.008** | **0.195±0.021** |
| | No GroupCLIP | 0.321±0.013 | 0.965±0.225 | 0.778±0.016 | 0.187±0.014 | 0.106±0.007 | 0.570±0.008 | 0.226±0.007 |
| | Autoencoder only | 0.280±0.013 | 0.846±0.175 | 0.793±0.013 | 0.195±0.012 | 0.111±0.006 | 0.574±0.006 | 0.240±0.012 |

## 6 DISCUSSION

This work introduces a multi-modal semi-supervised representation learning approach for weakly paired data leveraging a novel group contrastive loss (GroupCLIP) inside an *on-the-fly* backtranslating autoencoder. We also introduce combinatorial (with respect to OT aligners) benchmarking for multi-modal single cell perturbation alignment and cross-modal imputation. Empirical evaluation shows that GROOVE enables learning more useful latent representations for cell-level matching and downstream imputation. Ablations demonstrate our GroupCLIP is a crucial component for aligning the multi-modal representations and for good alignment performance. It is likly that GroupCLIP can have utility beyond just single cell data since weakly paired multi-modal data is ubiquitous in many domains (Yang et al., 2020; Mei et al., 2024; Sun et al., 2024). Our empirical results in Tables 1-4, powered by our robust evaluation framework, reveal that GroupCLIP-derived representations achieve particularly strong performance when paired with label-constrained OT methods (extended discussion on this topic in Appendix J.1). Furthermore, for the first time, we empirically show that optimal aligner choice varies across data modality pairs, methods, and shared variation percentages.

**Limitations.** While GROOVE demonstrates consistent performance across evaluated datasets, several limitations warrant discussion. First, like all contrastive approaches, our method is sensitive to hyperparameters; poor choices can lead to representation collapse or failure to leverage group structure. Second, our ablations reveal that backtranslation contributes are mixed, with greater effect in real data than in simulations, without pre-trained encoders, which were unavailable for our modalities. Third, the method assumes accurate perturbation labels, a standard assumption in single-cell perturbation screens where mislabeled samples are removed during quality control preprocessing. Label noise would corrupt any method using this as supervision signal, making this an experimental design consideration rather than a method-specific limitation. Fourth, neither GROOVE nor comparable methods have been evaluated under extreme class imbalance, which could significantly degrade performance. This should be more thoroughly evaluated in future work.

**Future directions.** Our work highlights several important directions for the community. First, developing more realistic simulation frameworks that incorporate variable proportions of shared versus modality-specific perturbation effects is critical, as the true regime in real datasets remains unknown. Second, there is a need for the community to establish consensus on evaluation priorities: whether sample matching or downstream tasks like imputation should serve as the primary performance criterion. Our results demonstrate that superior matching performance does not guarantee effective downstream imputation, indicating these objectives may require different methodological approaches. This prioritization decision is critical, as the optimal solutions for matching tasks likely differ from those that excel in downstream applications. Indeed our empirical results in both simulated and real data likely point to a tradeoff between matching and imputation when perturbation effects are not fully shared across modalities (extended discussion in Appendix J.2). Third, more sensitive imputation metrics with greater dynamic range are needed for robust perturbation prediction evaluation. Development of these metrics necessitates close partnership with biologists to define relevant research goals, facilitating the creation of evaluation frameworks that accurately capture performance on scientifically meaningful tasks. Fourth, to our knowledge, this is the first application of backtranslation architectures to single-cell data. While our ablations show some gains without pre-trained encoders, this framework provides a natural foundation for future multi-modal single-cell applications and should be further explored, particularly as suitable pre-trained models become available. Fifth, GroupCLIP and GROOVE do not make any strong assumptions limiting it to bimodal data and should generalize naturally to $> 2$ modalities, empirical validation of GROOVE and related methods in such setting remains valuable future work. Finally, systematic evaluation under extreme class imbalance would clarify method robustness boundaries.

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

# A  GROOVE EXTENDED DETAILS

## A.1  ON-THE-FLY BACKTRANSLATION

Each modality-specific encoder $g_\theta^{(m)}$ consists of a 3-layer MLP with batch normalization and ReLU activations, while each decoder $d_\theta^{(m)}$ uses a 2-layer MLP with an additional linear output layer (see next subsection). A shared linear projection (a coupling layer) $f_\theta$ connects all modality-specific encoders to their respective decoders (i.e. all modality specific embeddings are passed through this), giving the final encoder composition $f_\theta^{(m)} = f_\theta \circ g_\theta^{(m)}$.

In weakly paired single-cell data, each modality can be viewed as a different language with shared semantic information. We encourage cross-modally entangled representations by ensuring latents enable meaningful cross-modal translation through a self-backtranslation strategy. Let $\bar{m}$ denote the other modality. Given a sample's latent representation from modality $m$: $z_i^{(m)} = f_\theta \circ g_\theta^{(m)}(x_i^{(m)})$, we perform cross-modal translation in three steps:

1. **Cross-modal generation**: Using the decoder of modality $\bar{m}$ in inference mode, generate $x_i^{(m \to \bar{m})} = d_\theta^{(\bar{m})}(z_i^{(m)})$

2. **Re-encoding**: Switch to training mode and encode the generated sample: $z_i^{(m \to \bar{m})} = f_\theta \circ g_\theta^{(\bar{m})}(x_i^{(m \to \bar{m})})$

3. **Backtranslation**: Reconstruct the original modality: $x_i^{(m \to \bar{m} \to m)} = d_\theta^{(m)}(z_i^{(m \to \bar{m})})$

This *on-the-fly* (*on-the-fly*) process creates synthetic pseudo-paired samples within each mini-batch, contingent on cross-modally informative latent representations.

## A.2  ENCODER ARCHITECTURE AND REGULARIZATION

The first layer of the encoder projects each data modality from its native dimension $k^{(m)}$ to twice the size of the final embedding dimension ($2 \times d$). This design follows the recommendation of Samaran et al. (2024), who employ a variational-like encoder architecture where, for each $d$-dimensional embedding, the encoder outputs both a mean and log-variance parameter.

However, unlike standard variational autoencoders, we do not optimize the variational Evidence Lower BOund (ELBO), which combines reconstruction loss with Kullback-Leibler (KL) divergence. Instead, we minimize only the standard reconstruction loss (like a vanilla autoencoder). This design choice is motivated by prior work (Zhao et al., 2019), which demonstrated that KL divergence can conflict with reconstruction objectives and degrade downstream inference performance—a phenomenon we also observed in our internal analyses.

Our approach leverages the encoder's outputted parameters to define a Gaussian posterior distribution with diagonal covariance. Specifically, we interpret the $2d$-dimensional encoder output as mean $\boldsymbol{\mu}$ and log-variance $\log \boldsymbol{\sigma}^2$ parameters for a $d$-dimensional Gaussian distribution. During training, we sample from this distribution using the reparameterization trick (Kingma & Welling, 2013).

To ensure numerical stability and provide mild regularization, we add a small fixed constant ($10^{-4}$) to the diagonal covariance matrix during training only. This stochastic sampling mechanism serves three important purposes: (1) it introduces non-determinism in the encoder during training, (2) it prevents overcrowding of samples in the latent space by encouraging distributional rather than point estimates, and (3) it provides implicit regularization for the decoder, helping to prevent overfitting by requiring it to reconstruct from a distribution of latent codes rather than deterministic points.

During inference, the encoder operates deterministically using only the mean parameters $\boldsymbol{\mu}$.

## A.3  DECODER

The decoder takes the $d$-dimensional output from the shared projection layer $f_\theta$ and passes it through a 2-layer MLP with 1D batch normalization and ReLU activation and then a final linear layer top project the embeddings from $d$-dimensions back to the native $k^{(m)}$ features

## A.4 SIMILARITY KERNELS

The choice of similarity kernel significantly influences the quality and characteristics of learned representations. While cosine similarity remains the standard choice in most contrastive approaches, learning representations on a hypersphere (Radford et al., 2021; Chen et al., 2020), recent work demonstrates that a heavy tailed $t$-distribution parameterized can yield more expressive representations in Euclidean space (Böhm et al., 2022; Hu et al., 2022).

$$\text{cosine similarity}$$
$$\text{sim}(\boldsymbol{a}, \boldsymbol{b}) := \exp\left(\frac{\langle \boldsymbol{a}, \boldsymbol{b} \rangle}{\|\boldsymbol{a}\|_2^2 \cdot \|\boldsymbol{b}\|_2^2} \cdot \frac{1}{\tau}\right) \quad (7)$$

$$\text{$t$-distribution similarity}$$
$$\text{sim}(\boldsymbol{a}, \boldsymbol{b}) := \left[1 + \frac{\|\boldsymbol{a} - \boldsymbol{b}\|_2^2}{\tau\eta}\right]^{-\frac{\eta+1}{2}} \quad (8)$$

such that $\tau$ controls the temperature (bandwidth) parameter and $\eta$ is the degrees of freedom (which is set 1 for this work).

## A.5 MINI-BATCH BALANCED UNDERSAMPLING STRATEGY

To address possible class imbalance in our GroupCLIP framework, we implement a balanced under-sampling strategy that ensures equal representation across all classes within each batch. The algorithm first computes the instance count $n_l$ for each label $l$ in the first modality, assuming identical label distributions across both modalities. We then identify the minority class with the smallest instance count: $n_{\min} = \min_i(n_l)$.

Given an initial batch size $B$ and number of labels $L$, we compute the effective batch size using: $B_{\text{eff}} = B - (B \bmod L)$, where each labels contributes exactly $B_{\text{eff}}/L$ samples per batch. This ensures the batch size is evenly divisible by the number of labels. Importantly, our sampling strategy respects the minority labels constraint by never sampling more than $(B_{\text{eff}}/L \leq)n_{\min}$ instances from any label across the entire training process, preventing oversampling of the minority label.

For each batch, we randomly sample $B_{\text{eff}}/L$ instances from each label in both modalities independently, ensuring balanced representation while maintaining the weakly paired nature of the data. This approach prevents dominant labels from overwhelming the contrastive learning signal and ensures that all labels contribute equally to the learned representations, which is particularly important when dealing with imbalanced multi-modal datasets.

## A.6 ALGORITHM SKETCH

---

**Algorithm 1** GROOVE Training Procedure

---

**Require:** Modality-specific datasets $\mathcal{D}^{(1)}, \mathcal{D}^{(2)}$
**Require:** Encoders $g_\theta^{(1)}, g_\theta^{(2)}$, Decoders $d_\theta^{(1)}, d_\theta^{(2)}$, Shared Projection $f_\theta$
**Require:** Hyperparameters $\alpha$ (reconstruction), $\beta$ (GroupCLIP)
1: **for** while max training iteration is not reached **do**
2:     Sample balanced mini-batches $(\boldsymbol{x}^{(1)}, t^{(1)}) \sim \mathcal{D}^{(1)}, (\boldsymbol{x}^{(2)}, t^{(2)}) \sim \mathcal{D}^{(2)}$     ▷ See Section A.5
       **Step 1: Within-Modality Reconstruction and Contrastive Alignment**
3:     $\boldsymbol{z}^{(1)} \leftarrow f_\theta(g_\theta^{(1)}(\boldsymbol{x}^{(1)}))$
4:     $\boldsymbol{z}^{(2)} \leftarrow f_\theta(g_\theta^{(2)}(\boldsymbol{x}^{(2)}))$
5:     $\hat{\boldsymbol{x}}^{(1)} \leftarrow d_\theta^{(1)}(\boldsymbol{z}^{(1)})$
6:     $\hat{\boldsymbol{x}}^{(2)} \leftarrow d_\theta^{(2)}(\boldsymbol{z}^{(2)})$
7:     $\mathcal{L}_{\text{recon}} \leftarrow \sum_{m \in \{1,2\}} \text{MSE}(\boldsymbol{x}^{(m)}, \hat{\boldsymbol{x}}^{(m)})$
8:     $\mathcal{L}_{\text{GroupCLIP}} \leftarrow \frac{1}{2|\mathcal{D}_z^{(1)}|} \sum_{\boldsymbol{z}^{(1)} \in \mathcal{D}_z^{(1)}} \ell^{(1)} + \frac{1}{2|\mathcal{D}_z^{(2)}|} \sum_{\boldsymbol{z}^{(2)} \in \mathcal{D}_z^{(2)}} \ell^{(2)}$     ▷ See Equation 4
9:     $\mathcal{L}^{\text{step1}} \leftarrow \beta \cdot \mathcal{L}_{\text{recon}} + \alpha \cdot \mathcal{L}_{\text{GroupCLIP}}$
10:    Update parameters $\theta$ using gradients from $\mathcal{L}^{\text{step1}}$
       **Step 2: On-the-Fly Backtranslation**
11:    Set $g_\theta^{(1)}, g_\theta^{(2)}, d_\theta^{(1)}, d_\theta^{(2)}, f_\theta$ to Eval     ▷ Generate cross-modal samples w/ nograd
12:    $\boldsymbol{x}^{(1 \to 2)} \leftarrow d_\theta^{(2)} \circ f_\theta \circ g_\theta^{(1)}(\boldsymbol{x}^{(1)})$
13:    $\boldsymbol{x}^{(2 \to 1)} \leftarrow d_\theta^{(1)} \circ f_\theta \circ g_\theta^{(2)}(\boldsymbol{x}^{(2)})$
14:    Set $g_\theta^{(1)}, g_\theta^{(2)}, d_\theta^{(1)}, d_\theta^{(2)}, f_\theta$ to Train
15:    $\boldsymbol{z}^{(1 \to 2)} \leftarrow f_\theta(g_\theta^{(2)}(\boldsymbol{x}^{(1 \to 2)}))$     ▷ Re-encode translated samples
16:    $\boldsymbol{z}^{(2 \to 1)} \leftarrow f_\theta(g_\theta^{(1)}(\boldsymbol{x}^{(2 \to 1)}))$
17:    $\hat{\boldsymbol{x}}^{(1 \to 2 \to 1)} \leftarrow d_\theta^{(1)}(\boldsymbol{z}^{(1 \to 2)})$     ▷ Reconstruct original modality
18:    $\hat{\boldsymbol{x}}^{(2 \to 1 \to 2)} \leftarrow d_\theta^{(2)}(\boldsymbol{z}^{(2 \to 1)})$
19:    $\mathcal{L}_{\text{bt}} \leftarrow \frac{1}{2}[\text{MSE}(\boldsymbol{x}^{(1)}, \hat{\boldsymbol{x}}^{(1 \to 2 \to 1)}) + \text{MSE}(\boldsymbol{x}^{(2)}, \hat{\boldsymbol{x}}^{(2 \to 1 \to 2)})]$
20:    $\mathcal{L}^{\text{step2}} \leftarrow \beta \cdot \mathcal{L}_{\text{bt}}$
21:    Update parameters $\theta$ using gradients from $\mathcal{L}^{\text{step2}}$
22: **return** trained encoders $f_\theta^{(1)}, f_\theta^{(2)}$

---

## B  BASELINE DETAILS

Baseline Implementation Details All methods use identical encoder and decoder architectures[2] with a consistent latent embedding dimension of $d = 128$ across all experiments. For DAVAE, we adopt the previously reported optimal hyperparameter settings (Ryu et al., 2025). We evaluate GROOVE with both cosine and t-distribution (tdist) similarity kernels, setting $\alpha = 1.0, \tau = 0.2$, for all experiments and set $\beta = 0.1$ for single-cell data training. Note that we did not perform any rigorous hyperparameter exploration, we do not claim the reported results are the best-case performance of GROOVE for any of the evauated data sets.

We evaluated all representation learners in conjunction with five OT approaches: two standard methods and three label-constrained variants. The standard OT approaches include EOT and EG-WOT (Peyré et al., 2016; Kantorovich, 1960). The label constrained approaches include: labeledEOT, labeledEGWOT and labeledCOOT (co-optimal transport) (Ryu et al., 2025; Titouan et al., 2020). For all analyses, we use the default entropic regularizer settings from the `Perturb-OT` package[3].

## C  SIMULATION DETAILS

To simulate complex multi-modal cellular data, we define a probabilistic model that captures shared and modality-specific latent structures. The model begins by defining a shared latent space and two unshared, modality-specific latent spaces, which are then combined. The following perturbation types are used:

1. Shared perturbations: coordinately affect both modalities through the shared latent space with identical effect sizes and cell-specific penetrance values

2. Modality-specific perturbations: independently target each modality's unique dimensions with separate effect sizes and penetrance parameters

We initalize the simulations with the following settings:

- Latent dimensions: 10
- Shared variation proportions:
  - 100% shared: 10 shared, 0 unique dimensions
  - 80% shared: 8 shared, 2 unique dimensions per modality
  - 50% shared: 5 shared, 5 unique dimensions per modality
- Experimental design: 9 perturbations + 1 control condition
- Sample size: 100 cells per condition per modality
- Feature size: 1000 and 500 observed features for modalities 1 and 2, respectively

### C.1  GENERATIVE MODEL

We begin by defining the latent variables. A shared latent variable $Z$ is sampled coefficient-wise from a scaled standard normal distribution for $n$ total cells and $d_s$ shared dimensions:

$$Z_{ij} \sim \mathcal{N}(0, \text{scale}^2) \quad \text{for } i = 1, \ldots, n, \ j = 1, \ldots, d_s$$

where scale is a global latent signal strength (default 0.1).

Similarly, we define two unshared latent variables, $U_X$ and $U_Y$, with $d_u$ unshared dimensions:

$$U_{X,ij} \sim \mathcal{N}(0, \text{scale}^2), \quad U_{Y,ij} \sim \mathcal{N}(0, \text{scale}^2).$$

These are concatenated to form the full latent representations for each modality:

$$V_X = [Z \,\|\, U_X], \quad V_Y = [Z \,\|\, U_Y],$$

---

[2]PS does not require a decoder

[3]https://github.com/Genentech/Perturb-OT

with total dimensionality $d = d_s + d_u$.

Each modality is associated with a transformation matrix. For modality $X$ with $p_X$ features, the coefficients of $A_X \in \mathbb{R}^{d \times p_X}$ are sampled as

$$(A_X)_{jk} \sim \mathcal{N}(0, 1).$$

Similarly, for modality $Y$ with $p_Y$ features:

$$(A_Y)_{jk} \sim \mathcal{N}(0, 1).$$

Bias vectors $b_X \in \mathbb{R}^{p_X}$ and $b_Y \in \mathbb{R}^{p_Y}$ have coefficients

$$(b_X)_j \sim \mathcal{N}(0, 1), \quad (b_Y)_j \sim \mathcal{N}(0, 1).$$

Scaling factors $s_X \in \mathbb{R}^{p_X}$ and $s_Y \in \mathbb{R}^{p_Y}$ control feature variability:

$$(s_X)_j \sim \Gamma(1, 1), \quad (s_Y)_j \sim \Gamma(1, 1).$$

To account for modality-specific noise, perturbation parameters are defined coefficient-wise:

$$\mu_{X,j} \sim \mathcal{N}(0, 1), \quad \mu_{Y,j} \sim \mathcal{N}(0, 1),$$
$$\text{offsetsd}_{X,j} \sim \mathcal{N}(0, 1), \quad \text{offsetsd}_{Y,j} \sim \mathcal{N}(0, 1),$$
$$\sigma_{X,j} = \exp(-3.0 + \text{offsetsd}_{X,j}), \quad \sigma_{Y,j} = \exp(-3.0 + \text{offsetsd}_{Y,j}).$$

The noise for each cell $i$ and feature $j$ is then:

$$\xi_{X,ij} \sim \mathcal{N}(\mu_{X,j}, \sigma_{X,j}^2) \cdot \frac{\text{scale}}{\text{snr}}, \quad \xi_{Y,ij} \sim \mathcal{N}(\mu_{Y,j}, \sigma_{Y,j}^2) \cdot \frac{\text{scale}}{\text{snr}},$$

where snr is the signal-to-noise ratio (default 0.2).

We next incorporate $L$ perturbations across $L + 1$ conditions (including one control). Perturbations target shared or unshared latent dimensions in a cyclic manner.

For **shared perturbations**, the target dimension is

$$t_s(I) = ((I - 1) \bmod d_s).$$

Effect sizes are sampled as

$$|e_s(I)| \sim \max(3, \Gamma(1, 1)), \quad \text{sign}_s(I) \sim 2 \cdot \text{Bernoulli}(0.5) - 1,$$
$$e_s(I) = \text{sign}_s(I) \cdot |e_s(I)|.$$

Each cell $i$ has penetrance

$$q_{s,i} \sim \text{Beta}(1, 10).$$

For cells under perturbation $I$, the targeted latent dimension is shifted in both modalities:

$$v'_{X,it_s(I)} = v_{X,it_s(I)} + e_s(I)q_{s,i}, \quad v'_{Y,it_s(I)} = v_{Y,it_s(I)} + e_s(I)q_{s,i}.$$

For **modality-specific perturbations**, the target dimension is

$$t_u(I) = ((I - 1) \bmod d_u) + d_s.$$

Effect sizes are defined separately for each modality:

$$|e_{uX}(I)| \sim \max(3, \Gamma(1, 1)), \quad \text{sign}_{uX}(I) \sim 2 \cdot \text{Bernoulli}(0.5) - 1, \quad e_{uX}(I) = \text{sign}_{uX}(I)|e_{uX}(I)|,$$
$$|e_{uY}(I)| \sim \max(3, \Gamma(1, 1)), \quad \text{sign}_{uY}(I) \sim 2 \cdot \text{Bernoulli}(0.5) - 1, \quad e_{uY}(I) = \text{sign}_{uY}(I)|e_{uY}(I)|.$$

With penetrance

$$q_{uX,i} \sim \text{Beta}(1, 10), \quad q_{uY,i} \sim \text{Beta}(1, 10),$$

the latent variables are perturbed independently:

$$v''_{X,it_u(I)} = v'_{X,it_u(I)} + e_{uX}(I)q_{uX,i}, \quad v''_{Y,it_u(I)} = v'_{Y,it_u(I)} + e_{uY}(I)q_{uY,i}.$$

Let $V_X^{\text{pert}}$ and $V_Y^{\text{pert}}$ denote the final latent spaces. The observed data are generated as

$$X_{ij} = \left(((V_X^{\text{pert}} + \xi_X)A_X + b_X) \odot s_X\right)_{ij}, \quad Y_{ij} = \left(((V_Y^{\text{pert}} + \xi_Y)A_Y + b_Y) \odot s_Y\right)_{ij},$$

where $\odot$ denotes element-wise multiplication.

This model was implemented and simulations were generated using `Pyro`.

## D  PERTURB-MULTIOME PRE-PROCESSING

We downloaded the Perturb-Multiome data from (Martin-Rufino et al., 2025). We focused on the data at day 14 which showed the least cell type heterogeneity. To reduce the dimensionality and sparsity of the ATAC-seq data, we filtered ATAC-seq peaks to those measured in at least 1% of cells, and mapped peaks to the closest gene within with 100kb using the Gencode M38 annotation file. We, used the inverse document frequency and SVD to reduce the ATACseq to 256 components. For the RNA-seq data, we normalized the counts to 10,000 per cell and applied the log1p transformation. We filtered genes that are expressed in fewer than 1% of cells. We then subset to the top 512 highly variable genes using the Seurat approach with the scanpy implementation. We subset each perturbation to 128 cells per perturbation and split the data into RNA-seq and ATAC-seq for cross modal representation and prediction. We tested approaches accuracy at leveraging the ATAC-seq data to impute into the RNA-seq data.

## E  PERTURB-CITE-SEQ PRE-PROCESSING

We downloaded the Perturb-CITE-seq data from Franghei et al. (Frangieh et al., 2021). We focused on the IFNy condition and subset the data to that condition. The data was split into RNA-seq and CITE-seq for separate preprocessing. For RNA-seq, we normalized the total counts per cell to 10,000 and applied the log1p transformation. For both modalities we subtracted out the average expression of cells with control perturbations so all values in the gene or protein expression matrix are relative changes to the average control. To reduce the number of perturbations, we computed the energy distance for each target gene against the controls and subset to perturbations with at least 50 cells and an energy distance of 0.05. This resulted in 18 perturbations that were used for downstream tasks. Finally, we subset the gene expression to the top 500 highly variable genes using the Seurat method implemented in scanpy. We then tested approaches accuracy at leveraging the CITE-seq data to impute into the RNA-seq data.

## F   METRICS EXTENDED DETAILS

We evaluate cross-modal matching and prediction performance using eight complementary metrics that capture different aspects of alignment quality and distributional similarity.

**Trace Metric** Assuming the sample indices correspond to the true matching, we compute the average weight on correct matches, which is the normalized trace of the transport plan $\boldsymbol{T}$:

$$\text{Trace}(\boldsymbol{T}) = \frac{1}{n}\text{Tr}(\boldsymbol{T}) = \frac{1}{n}\sum_{i=1}^{n}\boldsymbol{T}_{ii} \tag{9}$$

The transport plan $\boldsymbol{T}$ is first row-normalized such that $\sum_j \boldsymbol{T}_{ij} = 1$ for all $i$. A uniformly random matching assigns $\boldsymbol{T}_{ij} = 1/n$ for each cell, yielding $\text{Trace}(\boldsymbol{T}) = 1/n$. Perfect matching yields $\text{Trace}(\boldsymbol{T}) = 1$.

**Barycentric FOSCTTM** We compute the Fraction Of Samples Closer Than the True Match using barycentric projection. Given matching matrix $\boldsymbol{T}$ and target data $\boldsymbol{X}^{(1)}$, we project to obtain $\hat{\boldsymbol{X}}^{(1)} = \boldsymbol{T}\boldsymbol{X}^{(1)}$. For each projected sample $\hat{\mathbf{x}}_i^{(1)}$, we compute the Euclidean distance to all samples in $\mathbf{X}^{(1)}$ and calculate the fraction of samples closer than the true match:

$$\text{FOSCTTM}(\boldsymbol{T}, \boldsymbol{X}^{(1)}) = \frac{1}{n}\sum_{i=1}^{n}\frac{1}{n-1}\sum_{j\neq i}\mathbf{1}\{d(\hat{\boldsymbol{x}}_i^{(1)}, \boldsymbol{x}_j^{(1)}) < d(\hat{\boldsymbol{x}}_i^{(1)}, \boldsymbol{x}_i^{(1)})\}, \tag{10}$$

where $d(\cdot, \cdot)$ denotes Euclidean distance. The final reported symmetric Barycentric FOSCTTM is an average over both both modalities: $0.5 \times (\text{FOSCTTM}(\boldsymbol{T}, \boldsymbol{X}^{(1)}) + \text{FOSCTTM}(\boldsymbol{T}^{\top}, \boldsymbol{X}^{(2)}))$

Lower values indicate better matching quality, with random matching expected to yield 0.5.

**Mean Squared Error (MSE)** For direct prediction evaluation, we compute the MSE between true samples $\mathbf{X}^{(1)}$ and predicted samples $\hat{\mathbf{X}}^{(1)}$:

$$\text{MSE} = \frac{1}{n}\sum_{i=1}^{n}\|\boldsymbol{x}_i^{(1)} - \hat{\boldsymbol{x}}_i^{(1)}\|_2^2. \tag{11}$$

We report the mean MSE across all features.

**1-Wasserstein Distance (WD)** To assess distributional similarity, we compute the 1-Wasserstein distance between true and predicted samples, averaged across features:

$$\text{WD} = \frac{1}{d}\sum_{j=1}^{d}W_1(\boldsymbol{X}_{:,j}^{(1)}, \hat{\boldsymbol{X}}_{:,j}^{(1)}), \tag{12}$$

where $W_1$ denotes the 1-Wasserstein distance between univariate distributions and $\boldsymbol{X}_{:,j}$ represents the $j$-th feature column.

**Cosine Similarity** We compute the average cosine similarity between corresponding true and predicted feature vectors:

$$\text{Cosine} = \frac{1}{d}\sum_{j=1}^{d}\frac{\boldsymbol{X}_{:,j}^{(1)} \cdot \hat{\boldsymbol{X}}_{:,j}^{(1)}}{\|\boldsymbol{X}_{:,j}^{(1)}\|_2\|\hat{\boldsymbol{X}}_{:,j}^{(1)}\|_2}, \tag{13}$$

where the dot product and norms are computed across samples for each feature.

**KNN-based Metrics** To evaluate neighborhood preservation, we construct $k$-nearest neighbor graphs for both true and predicted data using cosine similarity. Let $\boldsymbol{G}^{\text{true}}$ and $\boldsymbol{G}^{\text{pred}}$ denote the binary adjacency matrices of the respective KNN graphs (with $k = 10$). For each sample $i$, we treat $\boldsymbol{G}_{i,:}^{\text{true}}$ as ground truth labels and $\boldsymbol{G}_{i,:}^{\text{pred}}$ as predictions, then compute:

**KNN Recall**: The fraction of true neighbors correctly identified:

$$\text{KNN Recall} = \frac{1}{n}\sum_{i=1}^{n}\frac{\sum_j \boldsymbol{G}_{i,j}^{\text{true}}\boldsymbol{G}_{i,j}^{\text{pred}}}{\sum_j \boldsymbol{G}_{i,j}^{\text{true}}}. \tag{14}$$

**KNN Average Precision (KNN PR)**: The average precision score for each sample's neighborhood prediction:

$$\text{KNN PR} = \frac{1}{n} \sum_{i=1}^{n} \text{AP}(\boldsymbol{G}_{i,:}^{\text{true}}, \boldsymbol{G}_{i,:}^{\text{pred}}), \tag{15}$$

where AP denotes the average precision score.

**KNN ROC-AUC (KNN ROC)**: The area under the ROC curve for neighborhood prediction:

$$\text{KNN ROC} = \frac{1}{n} \sum_{i=1}^{n} \text{AUC}(\boldsymbol{G}_{i,:}^{\text{true}}, \boldsymbol{G}_{i,:}^{\text{pred}}), \tag{16}$$

where AUC denotes the area under the receiver operating characteristic curve.

# G  Simulations Extended Results

Table 6:  Statistical significance tests for cross-modal matching metrics comparing top-ranked GROOVE (Target) versus top-ranked non-GROOVE (Baseline) methods. One-sided paired t-tests and win rates computed across replicates, with test direction towards metric improvement. $\Delta$ = Target - Baseline mean difference; n.s. = p > 0.1.

| Shared Prop. | Metric | Target Method | Baseline Method | Avg. $\Delta$ | Win Rate (%) | p-value |
|---|---|---|---|---|---|---|
| 100% | Bary. FOSCTTM | GROOVE (cosine)+LabeledCOOT | DAVAE+LabeledCOOT | -0.039 | 100 | 8.50e-05 |
| | Trace | GROOVE (cosine)+LabeledCOOT | DAVAE+LabeledCOOT | 0.186 | 100 | 6.54e-06 |
| 80% | Bary. FOSCTTM | GROOVE (cosine)+LabeledCOOT | DAVAE+LabeledEOT | 0.034 | 0 | n.s. |
| | Trace | GROOVE (cosine)+LabeledCOOT | DAVAE+LabeledEOT | 0.072 | 100 | 1.29e-04 |
| 50% | Bary. FOSCTTM | GROOVE (cosine)+LabeledEOT | DAVAE+LabeledEOT | 0.000 | 60 | n.s. |
| | Trace | GROOVE (cosine)+LabeledEOT | DAVAE+LabeledEOT | -0.007 | 30 | n.s. |

Table 7:  Statistical significance tests for cross-modal imputation metrics comparing top-ranked GROOVE (Target) versus top-ranked non-GROOVE (Baseline) methods. One-sided paired t-tests and win rates computed across replicates, with test direction towards metric improvement. $\Delta$ = Target - Baseline mean difference; n.s. = p > 0.1.

| Shared Prop. | Metric | Target Method | Baseline Method | Avg. $\Delta$ | Win Rate (%) | p-value |
|---|---|---|---|---|---|---|
| 100% | Cos-sim | GROOVE (cosine)+LabeledCOOT | DAVAE+LabeledCOOT | 0.018 | 100 | 4.61e-05 |
| | KNN PR | GROOVE (cosine)+LabeledCOOT | DAVAE+LabeledCOOT | 0.028 | 100 | 1.11e-03 |
| | KNN ROC | GROOVE (cosine)+LabeledCOOT | DAVAE+LabeledCOOT | 0.017 | 100 | 2.17e-04 |
| | KNN Recall | GROOVE (cosine)+LabeledCOOT | DAVAE+LabeledCOOT | 0.033 | 100 | 2.17e-04 |
| | MSE | GROOVE (cosine)+LabeledCOOT | DAVAE+LabeledCOOT | -0.022 | 100 | 2.46e-06 |
| | WD | GROOVE (cosine)+LabeledCOOT | DAVAE+LabeledCOOT | -0.027 | 100 | 3.02e-06 |
| 80% | Cos-sim | GROOVE (cosine)+LabeledEOT | PS+LabeledEOT | 0.001 | 50 | n.s. |
| | KNN PR | GROOVE (cosine)+LabeledEOT | PS+LabeledEOT | 0.002 | 50 | n.s. |
| | KNN ROC | GROOVE (cosine)+LabeledEOT | PS+LabeledEOT | 0.001 | 60 | n.s. |
| | KNN Recall | GROOVE (cosine)+LabeledEOT | PS+LabeledEOT | 0.002 | 60 | n.s. |
| | MSE | GROOVE (cosine)+LabeledEOT | PS+LabeledEOT | 0.030 | 10 | n.s. |
| | WD | GROOVE (cosine)+LabeledEOT | PS+LabeledEOT | 0.001 | 40 | n.s. |
| 50% | Cos-sim | GROOVE (cosine)+LabeledEOT | PS+LabeledEOT | 0.005 | 90 | 1.35e-02 |
| | KNN PR | GROOVE (cosine)+LabeledEOT | PS+LabeledEOT | -0.001 | 50 | n.s. |
| | KNN ROC | GROOVE (cosine)+LabeledEOT | PS+LabeledEOT | -0.002 | 30 | n.s. |
| | KNN Recall | GROOVE (cosine)+LabeledEOT | PS+LabeledEOT | -0.003 | 30 | n.s. |
| | MSE | GROOVE (cosine)+LabeledEOT | PS+LabeledEOT | -0.003 | 50 | n.s. |
| | WD | GROOVE (cosine)+LabeledEOT | PS+LabeledEOT | 0.011 | 20 | n.s. |

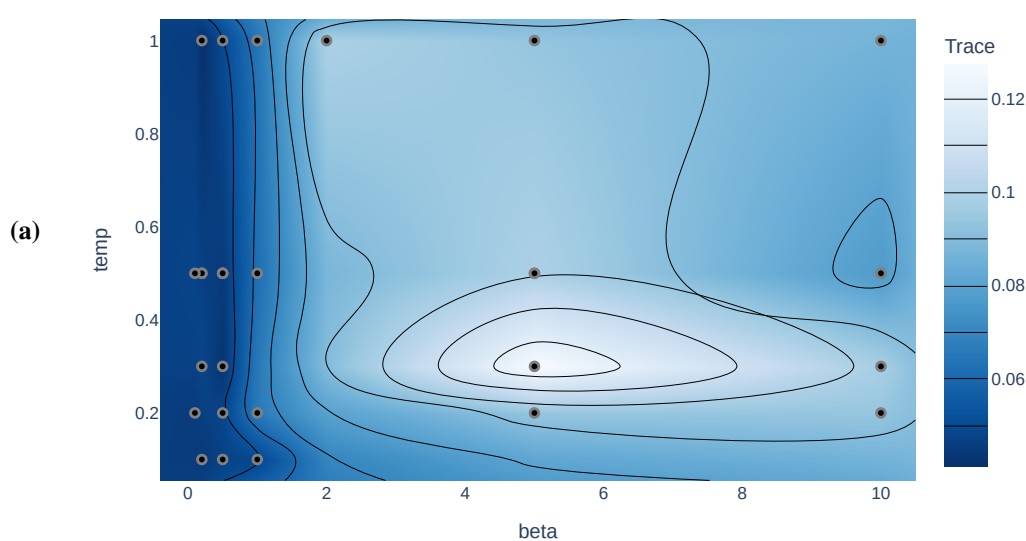

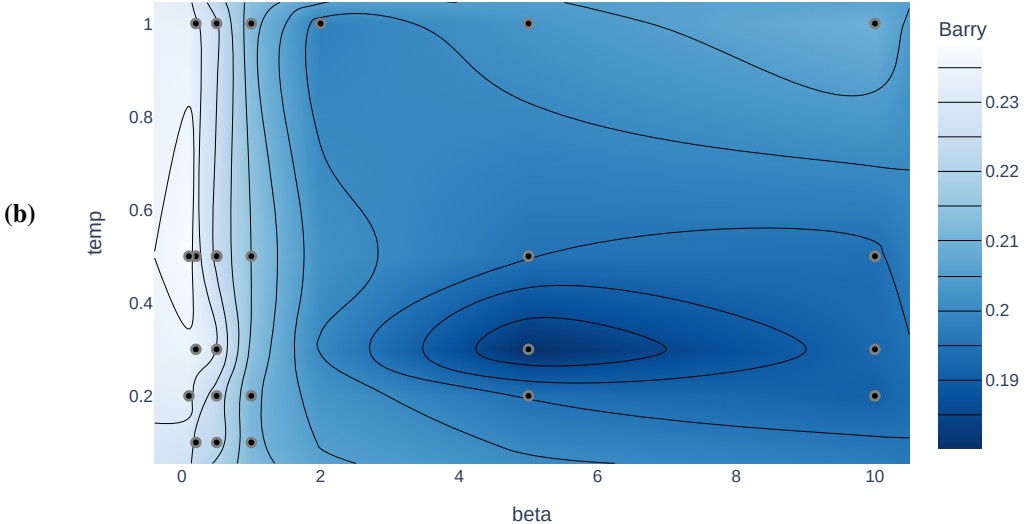

Figure 2: Hyperparameter sensitivity landscape for matching performance. Contour plots show average performance across 100%, 80%, and 50% shared variation settings for each combination of $\beta$ (x-axis) and $\tau$ (y-axis), profiled using Optuna-based hyperparameter search. $\alpha = 1$ for all analysis in this work. **(a)** Trace-based matching performance (higher is better). **(b)** Barycentric FOSCTTM (lower is better).

## H  PERTURB-CITE-SEQ EXTENDED RESULTS

Table 8: Matching performance metrics for top 10 method combinations in Perturb-CITE-seq dataset with 5-fold evaluation. SEs follow $\pm$; best in bold, second-best underlined.

| Method | Mean Rank | Trace | Bary. FOSCTTM |
|---|---|---|---|
| GROOVE (cosine)+ LabeledEOT | 3.0 | 0.039±0.002 | **0.381±0.004** |
| PS+ LabeledEOT | 3.0 | 0.039±0.002 | 0.383±0.002 |
| GROOVE (tdist)+ LabeledEGWOT | 3.5 | 0.033±0.001 | 0.365±0.001 |
| GROOVE (cosine)+ LabeledCOOT | 5.0 | **0.040±0.003** | 0.451±0.004 |
| GROOVE (cosine)+ LabeledEGWOT | 5.0 | 0.031±0.001 | 0.369±0.001 |
| GROOVE (tdist)+ LabeledEOT | 5.5 | 0.036±0.001 | 0.394±0.001 |
| DAVAE+ LabeledEGWOT | 7.5 | 0.024±0.000 | 0.383±0.003 |
| DAVAE+ LabeledEOT | 7.5 | 0.032±0.001 | 0.397±0.011 |
| GROOVE (tdist)+ LabeledCOOT | 7.5 | 0.036±0.002 | 0.452±0.004 |
| PS+ LabeledEGWOT | 9.0 | 0.023±0.000 | 0.389±0.001 |

Table 9: Matching performance metrics for top 10 method combinations in Perturb-CITE-seq dataset with leave one perturbation out evaluation. SEs follow $\pm$; best in bold.

| Method | Mean Rank | Trace | Bary. FOSCTTM |
|---|---|---|---|
| PS+ LabeledEOT | 2.5 | 0.008±0.001 | **0.487±0.004** |
| GROOVE (tdist)+ LabeledEOT | 4.5 | 0.008±0.002 | 0.490±0.003 |
| GROOVE (cosine)+ EGW | 6.0 | 0.008±0.001 | 0.491±0.003 |
| GROOVE (tdist)+ EOT | 6.0 | 0.007±0.001 | 0.485±0.003 |
| GROOVE (cosine)+ EOT | 6.5 | 0.007±0.001 | 0.489±0.003 |
| GROOVE (cosine)+ LabeledCOOT | 6.5 | **0.010±0.002** | 0.495±0.003 |
| PS+ EOT | 6.5 | 0.007±0.001 | 0.488±0.003 |
| GROOVE (cosine)+ LabeledEOT | 7.0 | 0.007±0.001 | **0.487±0.003** |
| DAVAE+ EOT | 7.0 | 0.008±0.002 | 0.491±0.003 |
| DAVAE+ LabeledCOOT | 7.5 | 0.008±0.002 | 0.495±0.002 |

Table 10: Imputation performance metrics for top 10 method combinations in Perturb-CITE-seq dataset with leave one perturbation out evaluation. SEs follow $\pm$; best in bold, second-best underlined, homogeneous metrics unannotated.

| Method | Mean Rank | MSE | Cos-sim | KNN Recall | KNN PR | KNN ROC | WD |
|---|---|---|---|---|---|---|---|
| GROOVE (cosine)+ LabeledEOT | 7.00 | 0.262±0.001 | 0.057±0.007 | 0.075±0.008 | 0.072±0.008 | 0.505±0.001 | 0.352±0.001 |
| GROOVE (tdist)+ LabeledEGWOT | 7.33 | 0.262±0.001 | 0.065±0.008 | 0.074±0.008 | 0.072±0.008 | 0.505±0.000 | 0.355±0.001 |
| DAVAE+ LabeledEOT | 7.83 | 0.262±0.001 | 0.082±0.008 | 0.074±0.008 | 0.073±0.008 | 0.504±0.001 | 0.354±0.000 |
| GROOVE (tdist)+ LabeledCOOT | 8.00 | 0.282±0.001 | 0.021±0.003 | 0.077±0.008 | 0.072±0.008 | 0.506±0.001 | 0.300±0.001 |
| PS+ LabeledEOT | 8.17 | 0.263±0.001 | 0.070±0.008 | 0.074±0.008 | 0.072±0.008 | 0.505±0.001 | 0.348±0.001 |
| GROOVE (cosine)+ EGW | 8.67 | 0.262±0.001 | 0.085±0.010 | 0.074±0.008 | 0.072±0.008 | 0.504±0.001 | 0.363±0.000 |
| DAVAE+ LabeledCOOT | 9.17 | 0.290±0.002 | 0.031±0.003 | 0.074±0.007 | 0.073±0.008 | 0.504±0.001 | **0.289±0.001** |
| GROOVE (cosine)+ LabeledEGWOT | 9.67 | 0.262±0.001 | 0.066±0.008 | 0.073±0.008 | 0.072±0.008 | 0.504±0.001 | 0.355±0.000 |
| GROOVE (tdist)+ LabeledEOT | 9.67 | 0.262±0.001 | 0.054±0.007 | 0.074±0.008 | 0.072±0.008 | 0.505±0.001 | 0.352±0.001 |
| DAVAE+ LabeledEGWOT | 10.33 | 0.262±0.001 | **0.088±0.009** | 0.071±0.008 | 0.073±0.008 | 0.502±0.001 | 0.354±0.001 |

Table 11: GROOVE ablation analysis performance metrics in Perturb-CITE-seq dataset. SEs follow ±; best in bold, second-best underlined, homogeneous metrics unannotated.

| Abation Type | Bary. FOSCTTM | MSE | Cos-sim | KNN Recall | KNN PR | KNN ROC | WD |
|---|---|---|---|---|---|---|---|
| GROOVE (cosine) | **0.368**±**0.005** | 0.261±0.001 | 0.047±0.002 | 0.019±0.001 | 0.017±0.000 | 0.502±0.000 | 0.353±0.000 |
| No GroupCLIP | 0.380±0.003 | 0.261±0.001 | 0.048±0.001 | 0.019±0.000 | 0.017±0.000 | 0.502±0.000 | 0.353±0.001 |
| Autoencoder only | 0.381±0.002 | 0.261±0.001 | 0.048±0.004 | 0.018±0.000 | 0.017±0.000 | 0.502±0.000 | 0.353±0.001 |

# I PERTURB-MULTIOME EXTENDED RESULTS

Table 12: Imputation performance metrics for top 10 method combinations in Perturb-Multiome dataset with 5-fold evaluation. SEs follow ±; best in bold, second-best underlined.

| Method | Mean Rank | MSE | Cos-sim | KNN Recall | KNN PR | KNN ROC | WD |
|---|---|---|---|---|---|---|---|
| GROOVE (cosine)+ EOT | 4.50 | **0.308**±**0.003** | 0.098±0.028 | 0.052±0.005 | 0.029±0.001 | 0.515±0.003 | 0.428±0.005 |
| GROOVE (cosine)+ LabeledEOT | 5.17 | 0.311±0.001 | 0.075±0.004 | **0.066**±**0.003** | **0.032**±**0.001** | **0.523**±**0.002** | 0.433±0.001 |
| PS+ EOT | 6.17 | **0.308**±**0.003** | **0.113**±**0.029** | 0.043±0.004 | 0.027±0.001 | 0.511±0.002 | 0.426±0.004 |
| GROOVE (tdist)+ LabeledEOT | 6.50 | 0.311±0.001 | 0.071±0.002 | **0.066**±**0.003** | **0.032**±**0.001** | **0.523**±**0.002** | 0.433±0.001 |
| GROOVE (tdist)+ EOT | 7.33 | 0.310±0.002 | 0.090±0.022 | 0.047±0.007 | 0.028±0.002 | 0.513±0.003 | 0.430±0.003 |
| GROOVE (cosine)+ LabeledCOOT | 7.50 | 0.355±0.023 | 0.091±0.054 | 0.045±0.005 | 0.028±0.001 | 0.512±0.002 | 0.323±0.008 |
| PS+ LabeledEOT | 7.50 | 0.311±0.001 | 0.070±0.002 | 0.058±0.003 | 0.030±0.001 | 0.518±0.001 | 0.433±0.001 |
| GROOVE (tdist)+ LabeledCOOT | 9.33 | 0.364±0.029 | 0.047±0.079 | 0.048±0.005 | 0.028±0.001 | 0.514±0.003 | **0.317**±**0.011** |
| PS+ LabeledCOOT | 9.50 | 0.334±0.006 | 0.041±0.025 | 0.048±0.005 | 0.028±0.001 | 0.514±0.002 | 0.379±0.001 |
| PS+ LabeledEGWOT | 10.00 | 0.311±0.001 | 0.075±0.003 | 0.037±0.006 | 0.026±0.001 | 0.508±0.003 | 0.430±0.001 |

Table 13: Matching performance metrics for top 10 method combinations in Perturb-Multiome dataset with leave one perturbation out evaluation. SEs follow ±; best in bold, second-best underlined.

| Method | Mean Rank | Trace | Bary. FOSCTTM |
|---|---|---|---|
| GROOVE (tdist)+ LabeledCOOT | 1.5 | **0.014**±**0.003** | 0.481±0.019 |
| GROOVE (tdist)+ LabeledEOT | 3.0 | 0.008±0.000 | **0.480**±**0.004** |
| GROOVE (cosine)+ EOT | 4.5 | 0.008±0.000 | 0.487±0.004 |
| DAVAE+ LabeledCOOT | 4.5 | 0.010±0.002 | 0.488±0.009 |
| GROOVE (cosine)+ LabeledEOT | 5.0 | 0.008±0.000 | 0.486±0.003 |
| GROOVE (tdist)+ EOT | 7.0 | 0.008±0.000 | 0.487±0.003 |
| GROOVE (tdist)+ LabeledEGWOT | 7.0 | 0.008±0.000 | 0.496±0.005 |
| GROOVE (cosine)+ LabeledCOOT | 7.5 | 0.011±0.003 | 0.500±0.018 |
| PS+ EOT | 9.5 | 0.008±0.000 | 0.497±0.002 |
| DAVAE+ LabeledEOT | 10.5 | 0.008±0.000 | 0.494±0.002 |

Table 14: Matching performance metrics for top 10 method combinations in Perturb-Multiome dataset with 5-fold evaluation. SEs follow $\pm$; best in bold, second-best underlined.

| Method | Mean Rank | Trace | Bary. FOSCTTM |
|---|---|---|---|
| GROOVE (cosine)+ LabeledCOOT | 2.0 | 0.048±0.009 | 0.431±0.028 |
| GROOVE (cosine)+ LabeledEOT | 2.5 | 0.041±0.001 | **0.427±0.008** |
| GROOVE (tdist)+ LabeledEOT | 4.5 | 0.040±0.000 | 0.441±0.003 |
| DAVAE+ LabeledCOOT | 6.0 | **0.052±0.010** | 0.470±0.025 |
| GROOVE (tdist)+ LabeledEGWOT | 6.0 | 0.040±0.000 | 0.448±0.003 |
| GROOVE (cosine)+ LabeledEGWOT | 6.5 | 0.039±0.001 | 0.446±0.003 |
| GROOVE (tdist)+ LabeledCOOT | 6.5 | 0.047±0.017 | 0.457±0.050 |
| PS+ LabeledEOT | 7.0 | 0.039±0.000 | 0.445±0.003 |
| DAVAE+ LabeledEGWOT | 8.5 | 0.039±0.001 | 0.448±0.003 |
| PS+ LabeledEGWOT | 8.5 | 0.039±0.001 | 0.447±0.003 |

# J EXTENDED DISCUSSION

## J.1 SYNERGY BETWEEN GROUPCLIP AND LABELED-CONSTRAINED OT

While GroupCLIP already leverages perturbation labels during representation learning, label-constrained OT methods provide complementary benefits at the alignment stage. Specifically, GroupCLIP operates at the group level during training, encouraging same-label samples to cluster together across modalities through *soft constraints*. Now consider the fact that perturbation effects are not always orthogonal. Under this common setting, the latent representations of cells with similar perturbation effects will be closer to each other (or more correlated) in the latent space. This is indeed the type of behavior one would desire from a useful/informative latent. Label-constrained OT on the otherhand sets a *hard constraint* across groups (see Ryu et al. (2025)). That is, it explicitly prevents cross-label matches while finding optimal pairings within each label group. This is a useful constraint when the objective is to match an individual cell within a perturbation. The empirical benefits of this synergy between our soft-constraint and Labelled OT's hard constraint can be quite strong: in Table 1 (100% shared), GROOVE with LabeledCOOT achieves Trace=0.856 versus markedly worse performance with unlabeled OT variants not even being in the top-5 (trace metrics strictly less than 5th highest combination).

## J.2 MATCHING VERSUS IMPUTATION TRADEOFF

Our empirical results reveal an important observation: superior matching performance does not necessarily translate to improved downstream imputation (Section 5). This phenomenon may reflect a tension between these two objectives that warrants further consideration.

Xi et al. (2024) argue that reconstruction losses force models to learn modality-specific noise, which is "counterproductive to matching." This claim rests on the assumption that all meaningful perturbation-induced variation is perfectly shared across modalities. However, biological reality is often more complex: perturbation effects frequently manifest differently across modalities, with some responses observable in only one modality (private or modality-specific variation) (Argelaguet et al., 2020; Lin & Zhang, 2023).

This can induce a tradeoff. For optimal matching, representations *can* capture only shared perturbation-relevant variation while discarding modality-specific information as "noise." Since in this task success is defined by recovering true instance-level pairs, i.e., the "real" co-measured cells. However, for imputation and prediction, the most useful training/inference samples are not necessarily the true paired cells, but rather the empirically most similar cells in the biologically relevant latent space (which might also factor in private variation). Fundamentally, perturbations induce similarity structures that supersedes instance-level pairing. Two unpaired cells subjected to the same perturbation

may exhibit greater functional similarity, and thus provide more informative training signal for imputation than a cell's true paired counterpart, particularly when factoring weak perturbation effects and technical measurement noise. A purely matching optimized representation, as Xi et al. (2024) does, discard this type of variation or stucture which would be helpful for downstream imputation.

GROOVE navigates this tradeoff through the inclusion of the reconstruction objective(s). The reconstruction and backtranslation losses (Equation 5) preserve sample-specific information, including modality-specific variation. While this may reduce pure matching performance by retaining what matching frameworks consider "noise," it provides empirically beneficial information for downstream imputation tasks. This explains why combinations with superior matching scores do not always achieve the best imputation performance (like in Tables 1 and 2). The consistency of this matching-imputation discordance also is need in real real datasets (Sections 5.1 and 5.2) suggests this represents a genuine phenomenon rather than an artifact of a particular dataset or experimental design. We believe this tradeoff warrants further theoretical investigation and empirical characterization, which we defer to future work.

