# OpenReview forum: "Group Contrastive Learning for Weakly Paired Multimodal Data"
_ICLR.cc/2026/Conference — Submitted to ICLR 2026_

### Official Review · Reviewer_JKiR · 2025-10-27

**Soundness:** 3
**Presentation:** 3
**Contribution:** 2
**Rating:** 4
**Confidence:** 4

**Summary:**

This paper introduces GROOVE, a semi-supervised representation learning method for weakly paired multimodal data. The approach integrates an on-the-fly backtranslating autoencoder with a group-level contrastive loss, GroupCLIP, which leverages shared labels to enforce cross-modal consistency within a shared latent space. The authors also propose a combinatorial evaluation framework that tests representation learners against various optimal transport (OT) alignment algorithms. The method is evaluated on cross-modal matching and imputation tasks across simulated and two real single-cell perturbation datasets.

**Strengths:**

* The paper addresses the important and difficult problem of "weakly paired" multimodal data. This is a common scenario in fields like single-cell biology, where destructive measurement techniques prevent collecting coupled omics data.

* The paper provides a robust evaluation framework, when compared to state of the art. By testing representation learners against various alignment algorithms, the authors decouple these two choices and provide a thorough assessment.

**Weaknesses:**

* While the problem addressed in this work is important, and the results seem solid and insightful to me, the novelty of the method appears  limited. On the one hand, it's simplicity is a strength, in my opinion, but the proposed methodology is not as preponderant as the effort the authors put in the experimental evaluation.

* Although the evaluation section is solid, in my opinion, it lacks several benchmarks established in the literature (see questions below).

**Questions:**

* The GROOVE method was mainly evaluated on ATAC, protein, and count data. How does it handle other kinds of modalities, such as images? The study by (Xi, 2024) [1] uses different benchmarks, including CITE-seq Data (NeurIPS 2021 challenge) and PerturbSeq/Single Cell Images, which provide more diversity in modality type. Would extra experiments on the datasets studied in [1] provide stronger empirical evidence of the method's generalizability? Is there time for the authors to assess the performance of the proposed method on such additional datasets?

* The ablation study in Table 5 is central to the claim that GroupCLIP is the key performance driver justifying the overall method performance. However, the "No GroupCLIP" model is effectively just the backtranslation framework, which the authors admit may be suboptimal without a pre-trained encoder. Can you elaborate more on how the ablation support the claim of GroupCLIP being so central to the performance of the proposed method?

* In [1], it is argued that any model with a reconstruction loss is forced to learn modality-specific noise, which is "counterproductive to matching". Could the authors discuss this claim in the context of GROOVE's autoencoder and backtranslation components? Furthermore, could this theoretical conflict explain your paper's own observation that "better matching performance does not guarantee optimal downstream task performance"? For instance, is it possible that the reconstruction-based losses (which the work in [1] argues are bad for matching) are actually necessary or beneficial for the downstream imputation task, thereby creating a trade-off between the two objectives?

[1] Xi, Johnny, et al. "Propensity score alignment of unpaired multimodal data." Neurips (2024)

---

> ### Author Response · Authors · 2025-11-21
>
> We sincerely thank the reviewer for the rigorous review and recognizing that our work addresses "an important and difficult problem" and for acknowledging that our results are "solid and insightful" with a "robust evaluation framework." We humbly address each of the concerns below.
>
> ## Re W1:
> We respectfully reframe this: GroupCLIP's parsimony is a core strength (as Reviewer uFXB also noted). Despite the pervasive nature of weakly-paired multimodal data in single-cell biology, there exists a fundamental gap in contrastive approaches leveraging group-level supervision without instance correspondence. GroupCLIP fills this gap. One could argue CLIP and SupCon are equally simple, yet their conceptual clarity enabled extensive adoption across domains. We next highlight our comprehensive evaluation framework as a contribution (see Re W1, Reviewer uFXB). Finally, this is the first backtranslation architecture proposed for single-cell data, providing a foundation for future multimodal applications.
>
>
> ## Re W2+Q1:
> We thank the reviewer for these insightful recommendations. We indeed considered these datasets when we started the project but chose alternatives for specific reasons:
>
> First, we used Perturb-CITE-seq instead of the CITE-seq dataset [1] because: a) CITE-seq lacks perturbations, [1] used cell types as labels. Cell type heterogeneity is typically much stronger than perturbation effects, making evaluation less realistic and easier. b) Both measure the same modalities (gene expression + protein), so we instead added the more recent Perturb-Multiome dataset [2] covering gene expression and chromatin accessibility for modality diversity.
>
> Second, we extensively searched for the PerturbSeq/Single Cell Images dataset from [1] but could not locate it, likely because it is proprietary or yet to be made public. If data has been made publicly available since this paper was written, we'd appreciate the link. Also, this is the same reason we could not use the primary evaluation dataset from [3]. *Noticing this issue in the current state of evaluation in the field, we made it a point to exclusively use publicly available datasets for reproducible benchmarking.* In the future, we intend to benchmark GROOVE on public PerturbSeq/Single Cell Image datasets as they become available.
>
> Finally, our simulations extend beyond [1]. Their simulations assume all perturbation-induced variation is shared between modalities. We relax this constraint, assessing performance across varying shared proportions. Thus, [1]'s simulation based evaluation represents only our 100% shared setting.
>
> ## Re Q2:
> We argue this comparison is appropriate for several reasons. First, we were not able to find a pre-trained multimodal encoder that is compatible across all our target modalities for this work, so evaluating backtranslation under the same constraints all methods face (no pre-training) is the fair comparison for our setting. Second, backtranslation and GroupCLIP served complementary and orthogonal objects. The former tries to maximize instance level cross-translation accuracy and consequently could help preserve within group variance, while the latter tries to enforce group level structure and coherence. A better (say a pre-trained) encoder should not diminish the contribution of GroupCLIP but improve the overall model performance. Finally, in Table 5 GROOVE with GroupCLIP outperforms both ablations without across all metrics with differences well beyond standard errors, demonstrating a consistent and meaningful performance improvement attributable to the GroupCLIP component.

---

> > ### Author Response · Authors · 2025-11-21
> >
> > ## Re Q3:
> > This is an excellent and insightful question. The reviewer has identified a tension between [1]’s theoretical prediction and our empirical observation (Lines 367-368): "better matching performance does not guarantee optimal downstream task performance."  First, [1] make a core assumption that all meaningful perturbation induced variation is shared across the two modalities; however, it's quite common in biology that some effects can only be seen in one modality and not others (private variation). Under this setting, a joint latent that is useful for imputation/reconstruction needs to be able capture both shared and modality specific perturbation effects. This can induce a tradeoff between matching and imputation performance. For matching, optimal representations can capture only shared perturbation-relevant variation while discarding modality-specific information. However, reconstruction requires predicting modality-specific features, which necessarily includes variation beyond what matching-optimized representations would discard as "noise." GROOVE implicitly makes this tradeoff with our reconstruction/backtranslation loss explicitly preserving some sample-specific information, including modality-specific variation, which may reduce pure matching performance but can be empirically helpful for imputation tasks.
> >
> > The evidence for this tradeoff replicates across simulation and real data. We agree that this observation needs further investigation and theoretical clarification, which we intend to do in future works. We have now added this in a new extended discussion Appendix J.2 and alluded to this in the main “Discussion” section.
> >
> > Finally, we appreciate the reviewer's detailed feedback and the opportunity to clarify our contributions. We hope our responses have adequately addressed the primary concerns. If the reviewer finds these responses adequate, we would appreciate a reconsideration of the overall recommendation. This process has meaningfully improved the manuscript and we're more than happy to address any additional questions and improve further!
> >
> > #### References
> > [1] Xi et al., 2024
> >
> > [2] Martin-Rufino et al., 2025
> >
> > [3] Ryu et al., 2025

---

> > > ### Comment · Reviewer_JKiR · 2025-11-24
> > > **Thank you for the rebuttal, no further questions on my side**
> > >
> > > Dear authors,
> > > this is a message to acknowledge and thank you for the rebuttal. I have read it and it positively addresses my main concerns.
> > >
> > > I have read the other reviews and your rebuttal for them. I now will wait for the discussion phase to begin.
> > >
> > > Thank you for your work!

---

> > > > ### Author Response · Authors · 2025-11-24
> > > >
> > > > Dear Reviewer JKiR,
> > > >
> > > > Thank you so much for your prompt review of our response and the really kind words! We're pleased that our rebuttal positively addresses your main concerns.
> > > >
> > > > It is our understanding (based on the email from the PCs) that discussion phase has already started and will continue on a rolling basis until December 2nd. If our responses and revised manuscript have adequately addressed your concerns and you now view the work favorably, we would be grateful if you could consider updating your score to reflect this.
> > > >
> > > > Of course, we remain ready to address any remaining questions or concerns you may have. And thank you again for your thorough review and constructive feedback throughout this process.

---

> > > > > ### Comment · Reviewer_JKiR · 2025-11-28
> > > > > **Thank you for the discussions**
> > > > >
> > > > > Dear authors,
> > > > > I've been monitoring the ongoing discussions stemming from other reviewers, your rebuttal, and their responses.
> > > > > Overall, it seems to me that the most frequent points all reviewers were arguing about are:
> > > > >
> > > > > * lack of statistical significance for the results
> > > > > * improvements needed for the ablation study
> > > > >
> > > > > In my opinion you have addressed those points, especially the second, with additional experiments, clarifications, or both. While any research work can be improved, I remain on my assessment: the methodological part of the paper is sufficiently solid, and the experimental part has been improved and represents a tangible contribution.
> > > > >
> > > > > For these reasons, I will raise slightly my score.

---

### Official Review · Reviewer_8e99 · 2025-10-29

**Soundness:** 2
**Presentation:** 3
**Contribution:** 2
**Rating:** 2
**Confidence:** 4

**Summary:**

- The paper tackles multimodal settings where examples from two modalities share only a group/label (e.g., perturbation ID) rather than instance-level pairs, and proposes a group-aware contrastive objective (“GroupCLIP”) that pulls together all samples from the same label across modalities while pushing apart different labels.
- The method combines this group-contrastive objective with a reconstruction pathway and on-the-fly “backtranslation” between modalities, yielding a shared embedding space that supports both matching and imputation.
- The paper argues that evaluating only instance-level matching is inadequate for weakly paired data, and introduces a combinatorial evaluation protocol that factors out aligner choice using labeled variants of entropic OT, GW-OT, and COOT.

**Strengths:**

- The paper addresses a realistic regime where modalities cannot be co-measured on the same cell, making group-level supervision both natural and necessary.
- The paper is well written and easy to follow
- The evaluation design separates representation learning from alignment by sweeping multiple labeled OT variants, which reduces confounding and yields a more credible comparison across methods.
- The ablations are clear and show that removing the group-contrastive term degrades performance consistently across metrics, which supports the central claim about the importance of group-aware contrast.

**Weaknesses:**

- CLIP’s web pairs are often weak and effectively many-to-one (e.g., many different dog images paired with near-identical captions), so large-scale CLIP training already approximates a group-level supervision regime rather than strict instance pairing. Thus it is important for the paper to show clear advantages in regimes where per-instance captions carry little unique information beyond a coarse label. The key claimed difference is that GROOVE does not need any per-instance pairing at all, whereas CLIP still relies on a text associated with each image; however, this difference is only compelling if the method outperforms strong CLIP-style baselines constructed to mimic weak pairs.
- The group-level objective risks collapsing within-label diversity because it contracts all samples of a label toward each other across modalities, which can be harmful when a perturbation has heterogeneous cellular responses; this risk is amplified by balanced per-label batching that repeatedly couples the same aggregate label sets.
- The backtranslation path adds complexity but appears to contribute less than the group-contrastive term in ablations, which raises questions about whether the extra module is necessary relative to a streamlined group-contrastive-only baseline with stronger decoders.
- The approach assumes label sets are perfectly aligned across modalities during training, but real datasets often contain partial, missing, or mis-specified label mappings; the paper does not evaluate robustness to label noise or missing labels, which is central to the weakly paired setting it targets.
- The proposed evaluation leans heavily on OT-based aligners, and while the authors sensibly sweep variants, the method’s ranking changes across datasets, which suggests sensitivity to the aligner choice. The paper does not analyze why particular aligners pair best with GROOVE or how to choose them reliably without oracle tuning.

**Questions:**

see weaknesses above

---

> ### Author Response · Authors · 2025-11-21
>
> We thank the reviewer for their feedback and for recognizing our work addresses "a realistic regime," that the paper is "well written and easy to follow," and that our "evaluation design separates representation learning from alignment" effectively. We address each concern below and respectfully clarify several points where there may have been misunderstandings about our problem setting.
>
> ## Re W1:
> We respectfully disagree with this characterization. There is a fundamental distinction: CLIP requires instance-level pairing where each image i has corresponding text i, enabling 1-to-1 training pairs. GROOVE operates under purely group-level supervision where no instance correspondence exists. Modality 1 contains N^(1) samples and Modality 2 contains N^(2) samples from different biological replicates sharing only group labels. Instance-level pairing is impossible, even weak pairing.
>
> The reviewer suggests showing "clear advantages...if the method outperforms strong CLIP-style baselines constructed to mimic weak pairs." However, constructing such baselines is impossible in our setting as there are no instance pairs to begin with, even weak ones. Creating "all-to-all" pairs for standard CLIP likely just results in semantically noisy or uninformative latents. GroupCLIP is the conceptually correct formulation for this setting, optimizing the "bag-level" alignment directly.
>
> Moreover, even if CLIP-based approaches could recapitulate group-level embeddings at scale (millions of samples), single-cell perturbation data exists in a limited-data regime (typically hundreds of cells per perturbation). Technical variation (batch effects, experimental noise) often dominates biological variation, corrupting implicit group structure. GroupCLIP addresses both limitations by explicitly leveraging group labels, making efficient use of limited samples while directly modeling perturbation-based structure. Thereby, addressing a significant gap in the literature, as other reviewers noted as well.
>
> ## Re W2:
> This is a thoughtful concern, but our empirical results demonstrate it does not occur. If GroupCLIP collapsed within-label variation, label-constrained OT would make random assignments (Trace = 1/N) since it relies on within-group variation for instance-level matching (also see Re Q2, Reviewer dmZz). Our results show this is not the case, GROOVE-derived latents match or outperform alternatives across simulations and real data.
>
> Additionally, the reconstruction loss implicitly preserves within-label variation, as label collapse would incur large reconstruction error. Finally, we undersample independently for each modality in each batch (Section A.5), ensuring different within-group samples are paired across training, preventing systematic coupling.
>
> ## Re W3:
> We have now added additional ablation results with real data (Peturb-Multiome) in Table 5 which does show backtranslation improving over the standard autoencoder. In light of these new results, we have significantly updated Section 5.3 and added possible reasons for this discrepancy and also highlight it further in a new “Limitations” paragraph in Section 6. Indeed, backtranslation does contribute less than GroupCLIP. But as noted in Section 5.3, we lacked pre-trained encoders (unavailable for multimodal single-cell data), which may underestimate its utility [1]. Lastly, this is the first backtranslation architecture proposed for single-cell data, establishing a foundation for future multimodal applications.

---

> > ### Author Response · Authors · 2025-11-21
> >
> > ## Re W4:
> > This raises an important point about label quality; however, we respectfully contextualize this concern within standard experimental practice. In single-cell perturbation screens, labels are experimentally assigned and validated through rigorous quality control, with mislabeled or contaminated samples removed during preprocessing, this is standard practice across all perturbation studies. Importantly, the assumption of accurate labels is not unique to GROOVE but is shared by all methods leveraging perturbation information [2,3]. Label noise would corrupt any method using this supervision signal, making this an experimental design consideration rather than a method-specific limitation. This has now been emphasized in the “Limitations” paragraph in Section 6. Regarding missing labels, GroupCLIP naturally accommodates scenarios where labels appear in only one modality (those samples simply have no cross-modal positives but retain negatives from other labels). Additionally, our leave-one-perturbation-out imputation benchmark (Section 5.2) directly evaluates this scenario, imputing perturbation effects present in only one modality, where GROOVE demonstrates competitive performance.
> >
> > ## Re W5:
> > We view this as a novel empirical finding, not a weakness. Our combinatorial framework reveals variation that prior work obscured by fixing either the learner [3] or aligner [2]. For the first time, we empirically show that optimal aligner choice varies across data modality pairs (Section 5.2, hypothesized but never demonstrated), methods, and shared variation percentages. These novel findings challenge the standard practice of fixing the aligner in benchmarks and should motivate development of more robust aligners. We’ve also now highlighted this in Section 6.
> >
> > While we cannot make strong empirical claims, we speculate the following: Label-constrained OT should be the default choice. For similar modalities (RNA+ATAC), use LabelledEOT (assumes shared feature space). For distinct modalities (RNA+protein), use LabelledCOOT (aligns geometric structures across different spaces).
> >
> > We hope our responses have satisfactorily addressed the concerns and clarified our contributions. If so, we would be grateful for reconsideration of the rating. We're more than happy to address any additional questions and improve further!
> >
> >
> > #### References
> > [1] Artetxe et al., 2017
> >
> > [2] Xi et al., 2024
> >
> > [3] Ryu et al., 2025

---

### Official Review · Reviewer_dmZz · 2025-10-29

**Soundness:** 3
**Presentation:** 3
**Contribution:** 3
**Rating:** 6
**Confidence:** 4

**Summary:**

This paper addresses an important problem in single-cell biology: learning representations from technically unpaired multi-modal data with shared functional labels such as perturbations (making it “weakly” paired). It introduces GroupCLIP, a novel contrastive loss extension for cross-modal representation alignment based on group-level supervision. This loss is combined with backtranslating autoencoders for higher-quality pseudo-pair generation. In an extensive evaluation with various optimal transport methods for matching, the method shows improved performance in cross-modal matching and imputation on small single-cell perturbation sets compared to baselines.

**Strengths:**

1) The paper tackles the important and practical challenge of learning from weakly paired multimodal data (where only group labels connect modalities), a common scenario in biological perturbation screens

2) The core contribution, the GroupCLIP loss effectively bridges the gap between cross-modal contrastive learning and uni-modal supervised contrastive learning for this specific weakly paired setting

3) The proposed method, GROOVE, outperforms the most comparable methods on real single-cell data for both matching and imputation

4) The paper includes both an ablation study showing that GroupCLIP is the main driver of the performance, as well as a thorough evaluation framework for all-against-all (learner and OT method) comparison.

5) The paper is generally well-written, clearly motivated, and provides substantial methodological detail (extensive appendices on architecture, sampling, baselines, and simulations)

**Weaknesses:**

1) Many of the performance differences reported in the simulation results (Table 1 Bary. FOSCTTM, Table 2) appear small and likely not statistically significant given the overlapping standard errors. The authors should provide a statistical test to show significant improvement.

2) The paper doesn't provide any analysis on a) sensitivity to hyperparameters alpha and beta that balance the GroupCLIP and reconstruction/backtranslation losses. It's unclear if the chosen values generalize or require dataset-specific tuning. b) the effectiveness of a balanced undersampling strategy is proposed, there's no analysis showing its effectiveness or the limits of imbalance the method can tolerate.

3) The discussion section focuses primarily on future directions for the community rather than critically analyzing the limitations of the proposed method itself (e.g., potential failure modes, hyperparameter sensitivity, unclear value of backtranslation).

4) The real-world datasets used are relatively small subsets derived after significant feature selection and focusing on specific experimental conditions. While understandable for computational reasons (OT scaling) and somewhat sufficient for demonstration on more homogeneous data, it leaves open the question of whether the method scales and performs well on larger and more heterogeneous datasets that are of interest for real-world screens.

5) The abstract's claim of "consistent outperformance in downstream cross-modal matching and imputation tasks" [lines 023f] is slightly overstated, as GROOVE did not significantly outperform the other methods on the simulations. Also, while GroupCLIP is novel, the overall GROOVE architecture relies heavily on adapting an existing backtranslating autoencoder framework from unsupervised machine translation. The ablation results (Table 5) also suggest the backtranslation component itself adds minimal value over a standard autoencoder in this setup, making the main effective innovation primarily the GroupCLIP loss. But since this is in an applications track and the results are outperforming alternative methods, this might be less important but it should be acknowledged in abstract/discussion.

6) While correctly identifying a gap for weakly paired supervised CLIP, the background could acknowledge existing supervised/semi-supervised CLIP extensions that use few perfect pairs (e.g., S-CLIP, SemiCLIP).

**Questions:**

1) The ablation description for "Autoencoder only" vs. "No GroupCLIP" could be clearer. Does "Autoencoder only" include GroupCLIP?

2) The results show mostly superior performance when using label-constrained OT methods. Could you briefly discuss why leveraging labels during the OT alignment step provides such a significant boost compared to standard OT, even after label-aware representation learning with GroupCLIP?

---

> ### Author Response · Authors · 2025-11-21
>
> We sincerely thank the reviewer for their thorough evaluation and positive assessment. We appreciate the recognition that our work "tackles an important and practical challenge," that "GroupCLIP effectively bridges the gap between cross-modal and uni-modal supervised contrastive learning," and that our paper is "generally well-written, clearly motivated, and provides substantial methodological detail." We address each concern below:
>
> ## Re W1:
> We appreciate this recommendation. We use overall ranking following established practice [1,2], as multiple metrics capture different aspects relevant to various downstream applications. We also report more comprehensive metrics than previous studies. We have now added one-sided paired t-tests and win rates in new Appendix G, comparing top-ranked GROOVE combinations against best alternatives in simulated settings.
>
> ## Re W2:
> For fairness, we evaluate GROOVE with fixed parameters (Lines 924-926: "we did not perform rigorous hyperparameter exploration"). Fine-tuning per dataset would’ve made our evaluation less rigorous. However, we have now conducted a hyperparameter search (using Optuna) on matching metrics to provide a quantitative evaluation of the hyperparameter sensitive landscape. Results are in new Appendix Figure 2. Next, regarding class imbalance: we employ undersampling by default since contrastive learning is sensitive to imbalance. Most experiments use balanced datasets (simulations and Perturb-Multiome). Only Perturb-CITE-seq is imbalanced, where GROOVE attains all top-5 imputation ranks, suggesting robustness. However, we cannot fully disentangle whether this stems from representation learning or the aligner. Systematically evaluating imbalance effects is an important direction for future work.
>
> ## Re W3:
> We appreciate this feedback. The reviewer is correct that our discussion emphasizes community directions but could better articulate our method's specific limitations. We have significantly restructured Section 6 and a clear “Limitations” paragraph.
>
> ## Re W4:
> Our real-world datasets match or exceed those in prior work [1,2] in sample size, features, and/or perturbation count. Feature selection is standard practice in single-cell biology due to sparsity and noise. Unlike previous studies, we prioritized publicly available data across multiple modality pairs, ensuring reproducibility despite constraining heterogeneity to what naturally exists in current datasets, which still includes substantial donor-level variation.
>
> Importantly, our simulation framework is the first to explicitly model shared and modality-specific perturbation effects. We demonstrate robust performance as shared variation decreases from 100% to 50%, substantially increasing heterogeneity. Finally, dataset size limitations stem from OT aligner scaling [2], not GROOVE (also see Reviewer uFXB Re Q2). Our GroupCLIP doesn't require all perturbations per batch, different perturbations can be sampled (uniformly) across batches; undersampling is only there to ensure within-batch balance.
>
>
> ## Re W5:
> We amended the abstract to: "performs on par with or outperforms"
>
> We have now added additional ablation results with real data (Peturb-Multiome) in Table 5 which does show backtranslation improving over the standard autoencoder. In light of these new results, we have significantly updated Section 5.3 and added possible reasons for this discrepancy and also highlight it further in a new “Limitations” paragraph in Section 6. As noted in Section 5.3, we lacked pre-trained encoders (unavailable for multimodal single-cell data), which may underestimate its utility. Lastly, this is the first backtranslation architecture proposed for single-cell data, establishing a foundation for future multimodal applications.
>
> ## Re W6:
> We thank the reviewer for highlighting these related works and apologize for the oversight. We have amended the “Contrastive representation learning” paragraph in Section 2 (Lines 103-106) to include these.

---

> ### Author Response · Authors · 2025-11-21
>
> ## Re Q1:
> Thank you for identifying this ambiguity. We now clarify this point in the main text by updating Lines 448-451.
>
> ## Re Q2:
> We thank the reviewer for this insightful question. Indeed, while GroupCLIP learns label-aware representations, label-constrained OT methods provide complementary benefits. GroupCLIP operates at the group level during training, encouraging same-label samples to cluster together across modalities through soft constraints. Now consider the fact that perturbation effects are not always orthogonal. Under this common setting the latent representations of cells with similar perturbation effects will be closer to each other (or more correlated) in the latent space. This is indeed the type of behavior one would desire from a useful/informative latent. Label-constrained OT approach on the other hand sets a hard constraint across groups (see [2]). That is, it explicitly prevents cross-label matches while finding optimal pairings within each label group. This is a useful constraint when the objective is to match an individual cell within a perturbation. The empirical benefits of this synergy between our soft-constraint and Labelled OT’s hard constraint can be quite strong: in Table 1 (100% shared), GROOVE with LabeledCOOT achieves Trace=0.856 versus markedly worse performance with unlabeled OT variants.
>
> We have now added this in a new extended discussion Appendix J.1 and alluded to this in the main “Discussion” section.
>
> Finally, we thank the reviewer for their thoughtful feedback, which has substantially improved our manuscript. We hope our responses adequately address the raised concerns and clarify the contributions of this work. If so, we respectfully request reconsideration of the rating. We remain available to address any further questions or concerns.
>
> #### References
> [1] Xi et al., 2024
>
> [2] Ryu et al., 2025
>
> [3] Artetxe et al., 2017

---

> > ### Comment · Reviewer_dmZz · 2025-11-25
> >
> > I thank the authors for their detailed response and for the additional analyses included in the revision. I appreciate the effort to address my concerns and recognize the theoretical merit of the proposed approach. However, several issues remain unresolved.
> >
> > Regarding Simulation Results (Re W1): While I understand the authors' point that different metrics capture different aspects of performance, the response does not alleviate the concern regarding the lack of sufficient significance, i.e. results in Supplementary Table 7 being largely insignificant.
> >
> > Regarding Hyperparameters (Re W2): I respectfully disagree with the premise that examining hyperparameter sensitivity makes an evaluation "less rigorous." On the contrary, understanding how performance degrades when parameters deviate from the optimal is crucial for establishing a method's robustness and practical utility for users who cannot perform extensive tuning. While an optuna search demonstrates that good parameters can be found, it does not serve as a sensitivity analysis to show the method's stability. The concern regarding whether the chosen parameters generalize or if the method is brittle remains.
> >
> > Regarding Dataset Scale and Preprocessing (Re W4): The authors state that drastic feature selection is "standard practice." While historically common due to computational limits, the single-cell machine learning field has largely moved toward utilizing high-dimensional data specifically because deep learning methods can exploit it. Validating a new deep learning method on such heavily filtered subsets limits the assessment of its utility in modern, large-scale perturbation screens where the high-dimensional feature space is a key advantage.
> >
> >
> > In summary, while GROOVE presents an interesting approach with GroupCLIP and addresses an important problem, the concerns regarding the statistical significance of the simulation results, the lack of clarity on hyperparameter robustness, and the evaluation on limited, highly processed datasets persist. The paper is sound but a marginal contribution in the current state. I encourage the authors to address these points in future revisions.

---

> ### Author Response · Authors · 2025-11-26
>
> We thank the reviewer for engaging in this discussion and their thoughtful follow-up!s We appreciate the acknowledgement that GROOVE "presents an interesting approach" and addresses "an important problem." We address each remaining concern below:
>
> **Re Simulation results**
>
> First, **prior works in this domain [1,2] did not report statistical significance tests, relying instead on relative ranking** to demonstrate improved performance. By introducing paired t-tests and win rates (Appendix G), we perform a more rigorous evaluation than the existing literature. Under this standard, GROOVE achieves statistically significant improvements (p<0.05) in the 100% shared setting. This is also the only simulated evaluation regime used by prior work (also see Re W2+Q1 Reviewer JKiR), which by itself constitutes an improvement in the state of the art, using their standards. We acknowledge that the 80% and 50% settings represent more challenging regimes (which we are the first to propose evaluation under) where all methods struggle, and we have been transparent about this in our manuscript.
>
> Second, and more importantly, the **real-data results provide compelling evidence of GROOVE's reliability**. In Perturb-CITE-seq (Table 3), GROOVE-based combinations occupy all top-5 ranks for imputation. In Perturb-Multiome evaluation (Table 4), GROOVE takes the top-3 ranks. This consistent top-tier performance across diverse real-world datasets, where biological complexity is fully present, demonstrates that GROOVE is meaningfully more consistent than existing approaches in practical settings where these methods would actually be deployed.
>
> **Re Hyperparameters**
>
> We apologize for our use of "less rigorous" as this was a stronger claim than intended. We would like to clarify two points. First, **we did not use Optuna to search for an optimum**, although that is its typical use case. Instead, we employed it for **hyperparameter sensitivity profiling and visualization**. As shown in Appendix Figure 2, all evaluated points lie on a systematic grid. We evaluated matching performance across 100%-50% shared proportion settings using fixed combinations: \tau = {0.1, 0.2, 0.3, 0.5, 1.0} and \beta = {0.1, 0.2, 1, 5, 10}. The resulting analysis is **effectively a sensitivity grid search**, not hyperparameter optimization. The contour plots in appendix Figure 2 directly visualize how performance degrades as parameters deviate from favorable regions, which we believe is the sensitivity analysis requested. If this is not the case, we kindly request the reviewer provide some more clarification.
>
> We apologize for the ambiguity and will re-clarify this in the figure Figure 2 caption.
>
> Second, **hyperparameter sensitivity is a general characteristic of contrastive learning**, not a limitation unique to GroupCLIP or GROOVE. Temperature parameters and loss weights require at least some tuning across virtually all contrastive methods (SimCLR[3], CLIP[4], SupCon[5]). We recommend users perform at least shallow hyperparameter exploration for new datasets, though **our defaults provide reasonable starting points** that achieved consistent and reasonable performance across evaluated datasets.
>
> **Re Datasets**
>
> We wholeheartedly agree that **future work should and will extend these approaches to larger datasets**.
>
> However, the **weakly-paired multimodal learning problem we address is fundamentally nascent**, with viable computational approaches emerging only in the last two years [1,2]. This in contrast to the mature unimodal or paired single-cell deep learning field the reviewer references, which has had over a decade of methodological development. **The datasets and preprocessing we employ represent the current commonly accepted standard** established by the only comparable prior works in this specific problem setting.
>
> Furthermore, our evaluation already represents a **substantial expansion** over prior work: we evaluate across multiple datasets (new simulations + two real, **public** datasets), multiple modality pairs, multiple evaluation paradigms (k-fold, LOPO), and an expansive range of metrics. Extending our combinatorial benchmarking framework to atlas-scale weakly-paired multimodal data would require evaluating over 15 method-aligner combinations across multiple metrics. This is a substantial undertaking that is not feasible within the rebuttal period, also in part due to external logistical constraints.

---

> > ### Author Response · Authors · 2025-11-26
> >
> > **Re Evaluating our contributions**
> >
> > Finally, we respectfully request that **the contributions of a body of work should be evaluated holistically** rather than on the basis of any single evaluation setting. As outlined in our response to other reviewers , **we make key contributions that span methodological contributions, novel evaluation frameworks and meaningful/novel empirical findings**. Namely:
> > - GroupCLIP: A novel group-level contrastive loss addressing a fundamental gap in the literature
> > - Comprehensive evaluation framework: Including novel simulations and combinatorial benchmarking that reveals previously unknown method-aligner dependencies
> > - Meaningful empirical findings: Including the first demonstration that optimal aligner choice varies across modality pairs and the identification of a matching-imputation tradeoff
> >
> > Taken together, we humbly argue these contributions do cross the threshold for meaningful scientific contributions. We hope our responses have adequately addressed the reviewer's remaining concerns.
> >
> > We believe that the reviewer's constructive criticism has quite meaningfully improved our manuscript and we remain available and eager for any further clarification that might help update their overall rating.
> >
> > #### References
> > [1] Xi et al. (NeurIPS, 2024)
> >
> > [2] Ryu et al. (AISTATS, 2025)
> >
> > [3] Chen et al. (ICML, 2020)
> >
> > [4] Radford, Kim, et al. (ICML, 2021)
> >
> > [5] Khosla, Teterwak, et al. (NeurIPS, 2020)

---

### Official Review · Reviewer_uFXB · 2025-10-31

**Soundness:** 3
**Presentation:** 2
**Contribution:** 2
**Rating:** 2
**Confidence:** 3

**Summary:**

The paper proposes GROOVE, a multimodal representation-learning approach for weakly paired data. In particular, GROOVE addresses challenges in single-cell perturbation analyses, where, while valuable, multimodal analysis with measurements from different modalities but the same cells can be infeasible to obtain. The proposed method combines a novel group-level contrastive loss (GroupCLIP), which leverages shared labels (across perturbations) to enforce consistency across modalities. It also leverages an on-the-fly backtranslating autoencoder to enforce well-mixed shared representations between modalities. The authors benchmark GROOVE against two standard baselines and compare representation learners across multiple optimal transport aligners. Experiments across simulated datasets (with varying degrees of modality sharing) and two real single-cell perturbation datasets show that  GROOVE (with an appropriate OT aligner) can lead to performance improvement in downstream cross-modal matching and imputation tasks. The ablation studies show that GroupCLIP is the key component for the performance gains.

**Strengths:**

- GroupCLIP is an extension that combines SupCon (using supervised class-labels for contrastive learning) and CLIP (for cross-modal alignment). Technically, it is a straightforward extension, but I find this simplicity a strength rather than a weakness.

- The motivation of the work is clear and addresses an important gap; multimodal methods that can leverage weakly paired data are crucial for biological applications where true paired measurements are experimentally infeasible

- A well-motivated experimental design evaluating different OT aligners.

**Weaknesses:**

- W1: The contributions of the paper are minimal. Besides GroupCLIP, the second contribution is the backtranslating autoencoder. This, however, doesn’t seem to have any positive effect on GROOVE’s performance.
- W2: The experimental analysis is limited to only two baselines. The authors discuss many more methods in the related work, but it’s unclear how these methods differ from GROOVE and why they weren’t chosen for benchmarking (for instance, Samaran et al, 2024)
- W3: Inconclusive findings wrt design choices of similarity metrics and OT aligners

**Questions:**

- The ablation study shows that removing GroupCLIP causes the largest performance drop, while backtranslation alone performs similarly to a standard autoencoder. Does this suggest that a simpler architecture employing GroupCLIP with a standard autoencoder would be equally effective? Have you evaluated this at different shared portion settings and real data?
- How does GROOVE scale computationally as the number of perturbations increases? The paper mentions ~20 perturbations, but this number could be much larger. Does the undersampling strategy become inefficient when there are many rare labels? Would this also affect the labeled OT aligners?
- How sensitive is GROOVE to the temperature parameter τ and the loss weights λ?

---

> ### Author Response · Authors · 2025-11-21
>
> We thank the reviewer for their constructive feedback and for acknowledging the clear motivation, the conceptual parsimony of GroupCLIP, and rigorous experimental design. We address each of the concerns below:
>
> ## Re W1:
> We respectfully clarify our three key contributions:
>
> 1. **GroupCLIP**: As recognized by other reviewers, we address a fundamental gap bridging CLIP and SupCon for weakly-paired settings. This is a novel problem formulation with significant biological implications where instance-level pairing is infeasible. We updated sections throughout the manuscript to clarify that GroupCLIP is the primary contribution of this work.
>
> 2. **Comprehensive evaluation framework**: We introduce: a) the first simulation framework systematically varying shared vs. modality-specific perturbation effects, a critical but previously unaddressed aspect of biological realism [1,2], and b) combinatorial benchmarking decoupling representation learning from alignment by evaluating all representation learners (including baselines) against multiple OT aligners. This results in 15 combinatorial baselines (3 learners and 5 OT aligners). This evaluation paradigm is itself a methodological contribution. This reveals that method rankings are sensitive to aligner choice (Tables 1-4), establishing evaluation protocols for this nascent field which lacks a rigorous evaluation framework.
>
> 3. **GROOVE architecture**: While ablations show backtranslation contributes less than GroupCLIP, this is itself a valuable empirical finding. However, we have now added additional ablation results with real data (Peturb-Multiome) in Table 5 which does show backtranslation improving over the standard autoencoder. Consequently, we have updated the text in Section 5.3 with possible reasons for this discrepancy and also highlight it further in a new “Limitations” paragraph in Section 6. Finally, this is the first backtranslation architecture proposed for single-cell data, establishing a foundation for future work.
> We have also revised the abstract to emphasize the evaluation framework more prominently and clarify the key contributions.
>
> ## Re W2:
> We appreciate the opportunity to clarify our baseline selection. We compare against all state-of-the-art methods capable of natively handling weakly-paired data:
>
> * **PS [2]**: The only method explicitly designed for cross-modal matching with group labels alone
>
> * **DAVAE**: Adapted from [5] to incorporate label supervision via linear probes
>
> This limited set actually demonstrates the novelty of our problem. Despite the biological importance of weakly-paired multimodal data, there is a striking gap in methods leveraging group-level supervision without instance correspondence.
> We exclude methods like Samaran et al. (2024)[6] because they require pre-defined feature correspondences (e.g., matching genes to peaks). This defeats the purpose of representation learning in disparate modalities (e.g., Imaging vs. RNA) where no such prior mapping exists. We also explicitly discuss this in Section 2 (Lines.146-149). It also is tracking a much more constrained problem. Furthermore, [6] cannot leverage perturbation labels, making it incompatible with our setting.
> Other related work methods require either: a) paired data (CLIP, CMC) or b) matched cells (MultiVI).
>
> We have amended Lines 266-267 in Section 4.2 to clarify this.
>
> ## Re W3:
> We respectfully disagree that our findings are "inconclusive." Rather, they reveal important dataset dependencies not systematically investigated before. For the first time, we empirically show that optimal aligner choice varies across data modality pairs (Section 5.2, hypothesized but never demonstrated), methods, and shared variation percentages. These novel findings challenge the standard practice of fixing the aligner in benchmarks and should motivate development of more robust aligners. Our evaluation also is the most comprehensive in this domain to date.

---

> > ### Author Response · Authors · 2025-11-21
> >
> > ## Re Q1:
> > Our ablation focused on relative component contributions while holding aligner and shared proportions fixed. As referenced in Re W1 above, we have extended our ablation analysis to the Perturb-Multiome dataset where we do indeed see an improvement from backtranslation over a standard autoencoder. As noted in Section 5.3, we lacked pre-trained encoders (unavailable for our single-cell data) [3], which may underestimate backtranslation's full utility. Please see the updated Section 5.3 for a full analysis. But additionally, backtranslation provides a principled framework for multimodal learning without instance-level pairing that should be further explored in single-cell data (see updated “Future directions” paragraph in Section 6).
> >
> > ## Re Q2:
> > Thank you for raising this point. Neither GroupCLIP nor GROOVE need all the perturbations to be sampled every batch. It’s acceptable to (uniformly) sample different sets of perturbations for each batch. The under-sampling is only to ensure balance within a single batch. Hence, GROOVE’s scaling is not limited by number of pertubations perturbation. Our within batch sampling strategy maintains efficiency by sampling exactly B_eff/L instances per label, where B_eff/L ≤ n_min, ensuring that scaling remains linear in the number of perturbations L for fixed B_eff. We never oversample minority classes (Section A.5), so the method remains efficient and robust when handling rare labels. Finally, while the OT evaluation step is computationally heavy because it is fit on the entire dataset, it is separate from our contributions and training GROOVE remains efficient and scalable.
> >
> > Our real-world datasets match or exceed those in prior work [4,7] in sample size, features, and/or perturbation count. The simulations are evaluated under 9 (+1 control) perturbations, the Perturb-Multiome analysis uses 20 (+1 control) transcription factor perturbations and Perturb-CITE-seq uses 19 (+1 control) genetic perturbations.
> >
> > ## Re Q3:
> > As with all deep learning approaches, hyperparameters affect performance. For fairness, we evaluated GROOVE with fixed parameters across datasets. As stated in Lines 924-926: "we did not perform rigorous hyperparameter exploration" and results are not best-case performance. However, we have now conducted a hyperparameter search (using Optuna) on matching metrics to provide a quantitative evaluation of the hyperparameter sensitive landscape. These results sweep the temperature parameter (\tau) and reconstruction/backtranslation (\beta) loss weights. Results are in new Appendix Figure 2 and discussed at the end of Section 5.3. We also humbly point out that we do not have a \lambda hyperparameter in GROOVE.
> >
> > Lastly, we hope our responses have addressed the reviewer's primary concerns and demonstrated the paper's contributions. If so, we would appreciate a reconsideration of the overall recommendation. We also believe the feedback so far has meaningfully improved the manuscript and we're more than happy to address any additional questions and improve further!
> >
> > #### References
> > [1] Argelaguet et al., 2020
> >
> > [2] Lin & Zhang, 2023
> >
> > [3] Artetxe et al., 2017
> >
> > [4] Xi et al., 2024
> >
> > [5] Ashuach et al., 2023
> >
> > [6] Samaran et al., 2024
> >
> > [7] Ryu et al., 2025

---

> > > ### Comment · Reviewer_uFXB · 2025-11-26
> > >
> > > I would like to thank the authors for their thoughtful response and extended analysis. I appreciate the effort and the discussion on my concerns. Overall, I can see the authors’ point of view, but I respectfully disagree with some of their points.
> > >
> > > Re:W1/Q1: I think my concerns largely remain. The authors argue that showing that “ backtranslation contributes less…is a valuable empirical finding”. Sure - but posing a design decision (ie, using backtranslation) that doesn’t contribute much (as shown by the empirical findings) as a contribution just by the fact that it is “a first” can be true for anything. I would argue that in this case, it would be more valuable to understand *why* it is underperforming, which might drive the efforts into better architecture or even a better understanding of the underlying data.  But this hasn't been tested. I do appreciate the additional experiments. But I still fail to see more practical evidence of “backtranslation improving over the standard autoencoder”. It does, but insignificantly, and this is consistent across the metrics (for example, the barycentric fraction indicates near random assignments). The synthetic data experiments could be a good opportunity to test these effects under different conditions.
> > >
> > > Re:W3: I would also respectfully disagree with this argument. My comment on the choice of optimal OT being  “inconclusive” is based on the results presented in the tables, which points to the need of more evidence. Your claims in the paper (L415-417) and in the rebuttal do not address this. Your conclusion is that the choice of the OT is data dependent, which would mean tuning/evaluating per task (which is another way of saying “inconclusive”).
> > >
> > > Re:Q2/Q3: I appreciate the discussion and additional HP experiments (also thanks for the correction re:$\lambda$ ; indeed I was referring to the parameters in the loss, and esp. $\beta$). That said, I do not agree with the authors’ argument of not performing it in the first place (as given in the response to `dmZz`)
> > >
> > > As stated in my original review, this is an interesting work. However, IMO, the weaknesses still outweigh the strengths, and haven’t been fully addressed in the response. I will increase the score to reflect the authors’ clarifications to some of the points (and additional ablations). But I remain that in the current form, the paper will benefit from further revisions and extended analysis of some of the main components of GROOVE (beyond GroupCLIP).

---

> > > > ### Author Response · Authors · 2025-12-01
> > > >
> > > > We sincerely thank the reviewer for their continued engagement and for increasing the score to reflect our clarifications and additional experiments. We really appreciate the constructive dialogue and address the remaining concerns below.
> > > >
> > > > **Re W1/Q1: Backtranslation**
> > > >
> > > > We appreciate the reviewer's push for deeper understanding and acknowledge that **GroupCLIP is unambiguously our primary methodological contribution**. We have been transparent about this throughout the manuscript and rebuttals.
> > > >
> > > > However, we respectfully offer several points of clarification. **First, regarding "why" backtranslation shows mixed results**. We provided explicit hypotheses in Section 5.3 and the “Limitations” paragraph. The original on-the-fly backtranslation framework [1] explicitly states that **pre-trained cross-lingual embeddings are essential “in the early stages of training”** to bootstrap the system into producing meaningful translations. Without such initialization, the authors note backtranslation can signifncatly underperform its full potential. No such pre-trained multimodal encoders exist, *yet*, for single-cell data. This architectural prerequisite **likely explains the attenuated gains**. We believe this explanation *is* an understanding of why the component underperforms, and it directly motivates future work on pre-trained multimodal encoders for single-cell data.
> > > >
> > > > **Second, regarding practical evidence**. We respectfully highlight that in **real data (Perturb-Multiome, Table 5)**, backtranslation *does* improve over the standard autoencoder across most metrics. While improvements are modest, they are **consistent in direction**. The reviewer notes the Barycentric FOSCTTM indicates near-random assignments, but notice that it *is* random without backtranslation. Additionally, this setting specifically evaluates matching quality of the *ablated* models (without GroupCLIP), which we expect to perform poorly and shows why GroupCLIP is essential.
> > > >
> > > > **Third, regarding the "first" framing**. We acknowledge the reviewer's point that novelty alone is insufficient. However, **establishing that a technique does not transfer straightforwardly to a new domain is scientifically valuable**. This negative result prevents future researchers from assuming backtranslation will yield similar gains as in NLP and directs efforts toward possible conditions (pre-trained encoders) that enable its success.
> > > >
> > > > Finally, we emphasize that GROOVE's architecture is **modular and practitioners can omit backtranslation and retain the full benefits of GroupCLIP**. This flexibility is a practical strength.
> > > >
> > > > **Re W3: OT Aligner Choice**
> > > >
> > > > We appreciate the reviewer's clarification and humbly offer a reframing of the term "inconclusive." The reviewer states that "data-dependent" is "another way of saying inconclusive." We respectfully disagree with this equivalence. **"Inconclusive" implies we failed to learn anything; "data-dependent" describes a discovered empirical regularity that advances the field's understanding**. Prior work in this domain **fixed either the representation learner [2] or the OT aligner [3]**, making it impossible to detect these dependencies. Our combinatorial framework is the first to systematically reveal that (a) method rankings change across datasets and modality pairs, and (b) label-constrained OT consistently outperforms unconstrained variants. This is **more informative than prior work** [2,3], which offered no such guidance.
> > > >
> > > > We acknowledge that **no single aligner uniformly dominates** but this is a property of the problem, not a limitation of our evaluation. Discovering this property is itself a contribution that challenges the standard practice of fixing aligners in benchmarking.

---

> > > > > ### Author Response · Authors · 2025-12-01
> > > > >
> > > > > **Re Q2/Q3: Hyperparameter Analysis**
> > > > >
> > > > > We appreciate the reviewer's acknowledgment of our additional experiments. We would like to clarify a potential misunderstanding regarding our position.
> > > > >
> > > > > First, **we did not use Optuna to search for an optimum**, although that is its typical use case. Instead, we employed it for **hyperparameter sensitivity profiling and visualization**. As shown in Appendix Figure 2, all evaluated points lie on a systematic grid. We evaluated matching performance across 100%-50% shared proportion settings using fixed combinations: \tau = {0.1, 0.2, 0.3, 0.5, 1.0} and \beta = {0.1, 0.2, 1, 5, 10}. The resulting analysis is **effectively a sensitivity grid search**, not hyperparameter optimization. The contour plots in appendix Figure 2 directly visualize how performance degrades as parameters deviate from favorable regions, which we believe is the sensitivity analysis requested. We believe this constitutes a meaningful sensitivity analysis.
> > > > >
> > > > > Our original statement about "not performing rigorous hyperparameter exploration" referred to **not optimizing hyperparameters separately for each dataset**, which we maintain is a valid evaluation design choice. Optimizing per-dataset could indeed inflate our reported performance relative to baselines that can't be tuned like PS.
> > > > >
> > > > > We hope our responses have further clarified our contributions and addressed the remaining concerns.
> > > > >
> > > > > **We also acknowledge that this rebuttal was finalized shortly before the decision to close the discussion period. We have decided to submit it nonetheless to ensure a complete response to the points raised.**
> > > > >
> > > > > #### References
> > > > >
> > > > > [1] Artetxe et al. (ICLR, 2018)
> > > > >
> > > > > [2] Ryu et al. (AISTATS, 2025)
> > > > >
> > > > > [3] Xi et al. (NeurIPS, 2024)

---

### Author Response · Authors · 2025-12-02
**Discussion Summary for New AC (1/3)**

Given the circumstances surrounding the OpenReview incident and the decision to revert scores, we provide a summary of the reviews and discussion for the new area chair. We hope this will helps the contextuatize the work and changes made during the discussion period, independent of the orginal rating. We thank the reviewers for their constructive feedback, which substantially improved the submission. All changes are marked in red in the revised PDF.

Our manuscript received discussion from three of four reviewers prior to closure. **Reviewer JKiR explicitly raised their score**, stating: *"I've been monitoring the ongoing discussions... Overall, it seems to me that the most frequent points all reviewers were arguing about are: lack of statistical significance for the results, improvements needed for the ablation study. In my opinion you have addressed those points, especially the second, with additional experiments, clarifications, or both... the methodological part of the paper is sufficiently solid, and the experimental part has been improved and represents a tangible contribution"*. **Reviewer uFXB also increased their score** to reflect our clarifications and additional ablations. Lastly, the discussion period unforturtaly ended **before we could hear back from Reviewer dmZz** on our second-round rebuttal addressing their remaining concerns.
***
# Strengths
Strengths noted consistently across reviewers include:

1. **GroupCLIP addresses an important gap**: Reviewers acknowledged we tackle "an important and practical challenge" (dmZz), "an important and difficult problem" (JKiR), and address "a realistic regime" (8e99) where instance-level pairing is infeasible. The “theoretical merit of the proposed approach” (dmZz) was recognized and simplicity of GroupCLIP was noted as "a strength rather than a weakness" (uFXB) and.

2. **Robust evaluation framework**: The combinatorial evaluation separating representation learning from alignment was praised as "well-motivated" (uFXB) and providing "more credible comparison across methods" (8e99)

3. **Clear presentation**: The paper is "well-written, clearly motivated, and provides substantial methodological detail" (dmZz) and "easy to follow" (8e99)

4. **Solid empirical results**: Results are "solid and insightful" (JKiR) with GROOVE outperforming comparable methods on real single-cell data (dmZz)

---

> ### Author Response · Authors · 2025-12-02
> **Discussion Summary for New AC (2/3)**
>
> # Weaknesses Addressed During Discussion
> We addressed the primary concerns raised by reviewers through additional experiments and clarifications:
>
> **1. Backtranslation Contribution** (uFXB, dmZz, 8e99, JKiR)
>
> We clarified that **GroupCLIP is unambiguously our primary contribution**. However, we added real-data ablations (Table 5) showing **backtranslation does improve over standard autoencoders in Perturb-Multiome**. We next provided **explicit hypotheses for mixed results**: the original on-the-fly backtranslation framework requires pre-trained encoders "in the early stages of training" [1], which do not yet exist for single-cell data. This explanation identifies *why* the component underperforms and motivates future work. Importantly, GROOVE's architecture is modular; practitioners can omit backtranslation and retain full GroupCLIP benefits.
>
> **2. Hyperparameter Sensitivity** (uFXB, dmZz)
>
> We conducted a systematic grid search across \tau = {0.1, 0.2, 0.3, 0.5, 1.0} and \beta = {0.1, 0.2, 1, 5, 10}. Appendix Figure 2 presents contour plots visualizing how performance changes across different hyperparameter combination. **This constitutes a sensitivity analysis, not hyperparameter optimization**. We also note that hyperparameter sensitivity is a general characteristic of contrastive learning, not a limitation unique to GroupCLIP or GROOVE.
>
> **3. OT Aligner Choice Being "Inconclusive"** (uFXB)
>
> We respectfully reframed this concern: "Inconclusive" implies we failed to learn anything; "data-dependent" describes a discovered empirical regularity. Prior work fixed either the representation learner [2] or aligner [3], making it impossible to detect these dependencies. Our combinatorial framework is the first to reveal that (a) method rankings change across datasets and modality pairs, and (b) label-constrained OT consistently outperforms unconstrained variants. **This advances the field's understanding beyond prior work**.
>
> **4. Significant Improvement** (dmZz, uFXB)
>
> We added paired t-tests and win rates (new Appendix G). **GROOVE achieves statistically significant improvements (p<0.05) in** the 100% shared setting. This is the **standard evaluation regime used by prior work [2,3]** and by itself constitutes an **improvement in the state-of-the-art**. The more challenging 80% and 50% settings, where all methods struggle, represent our novel contribution to evaluation; no prior baselines exist for these regimes. We also emphasized that **real-data results provide compelling evidence**: GROOVE occupies all top-5 ranks for imputation in Perturb-CITE-seq (Table 3) and top-3 ranks in Perturb-Multiome (Table 4).
>
> **5. Datasets** (dmZz)
>
> Our **datasets match or exceed prior work [2,3]** in sample size, features, and perturbation count. We **prioritized publicly available data for reproducibility** which also **addresses a gap** we identified where key evaluation datasets from prior work remain proprietary.
>
> **6. Baseline Selection** (uFXB)
>
> We clarified that **we compare against all methods capable of natively handling weakly-paired data without pre-specified feature correspondences**. Methods like Samaran et al. [4] require pre-defined feature mappings between modalities, defeating the purpose of representation learning for disparate modalities. This **limited baseline set actually demonstrates the novelty of our problem formulation and need for for approaches like ours.

---

> ### Author Response · Authors · 2025-12-02
> **Discussion Summary for New AC (3/3)**
>
> # Summary of Contributions
> 1. **GroupCLIP**: A novel group-level contrastive loss addressing a fundamental gap in the literature.
>
> 2. **Comprehensive evaluation framework**: Including novel simulations varying shared vs. modality-specific effects (previously unaddressed) and combinatorial benchmarking revealing method-aligner dependencies
>
> 3. **GROOVE architecture**: First backtranslation framework for single-cell data, providing foundation for future multimodal applications
>
> 4. **Meaningful empirical findings**: First demonstration that optimal aligner choice varies across modality pairs, shared proportion settings and datasets; identification of a matching-imputation tradeoff; synergy between GroupCLIP and label-constrained OT
> ***
> **The engaged reviewers' improved sentiment emerged through constructive dialogue, for which are very grateful**. Reviewer JKiR explicitly stated our rebuttals "positively addressed [their] main concerns." Reviewer uFXB acknowledged our "clarifications" and "additional ablations" warranted raising their score. We hope the area chair is similarly convinced by the merits of the rebuttal and the complete discussion below.
>
> **We respectfully argue that every primary concern raised was addressable through additional experiments or clarifications, and that the paper makes clear methodological and empirical contributions to a nascent but important problem.**
>
> _**Finally, we are grateful for the area chair's time in carefully considering our submission under these unusual circumstances.**_
>
> #### References
>
> [1] Artetxe et al. (ICLR, 2018)
>
> [2] Ryu et al. (AISTATS, 2025)
>
> [3] Xi et al. (NeurIPS, 2024)
>
> [4] Samaran et al. (Nat. Comm., 2024)

---

### Meta-Review · Area_Chair_Yga7 · 2026-01-07

**Summary:**

main concerns from reviewers are summarized below:

Limited contributions: backtranslating autoencoder not clear to help, lack of acknowledgement to existing supervised/semi-supervised CLIP extensions that use few perfect pairs (e.g., S-CLIP, SemiCLIP).

Experimental concerns: insufficient baselines, ablation studies don't justify improvement from key components, limited performance gains, no statistical tests, hyperparameter sensitivity analysis, no limitations of proposed method, datasets used are too small, does not evaluate robustness to label noise or missing labels central to the weakly paired setting.

**Reviewer Concerns:**

limited contributions

--> seems most reviewers still doubt the importance of backtranslating autoencoder component after the rebuttal. concerns remain about the increased complexity and lack of evidence showing its necessity.

experimental rigor

--> seems the authors addressed some of the concerns including more ablations, experiments, etc... but reviewers also point out the lack of rigor in not performing such in-depth analysis in the first place, and questioned other soundsness concerns in the paper.

experimental results

--> authors added statistical significance for the results, and generally improved the ablation study. other conceptual questions were also addressed

**Reviewer Scores:**

JKiR who gave 4 might increase to 6.

uFXB who gave a 2 engaged in a very detailed back and forth with the authors, but significant concerns remain on the importance of backtranslation component, lack of conclusive evidence for the importance of OT, and poor soundness of experimental rigor and analysis. probably would have increased to 4 but still lean negative.

8e99 who gave a 2 did not engage in discussion but raised many important concerns (also overlapping with other reviewers) regarding importance of backtranslation path, lack of robustness studies, and experimental concerns. The authors addressed some of these but not all, they might increase score to 4, but likely still negative.

---

### Decision · Program_Chairs · 2026-01-26

Reject